# One-Inlier is First: Towards Efficient Position Encoding for Point Cloud Registration

**Fan Yang**  **Lin Guo**  **Zhi Chen**  **Wenbing Tao**[*]
School of Artificial Intelligence and Automation
Huazhong University of Science and Technology, Wuhan 430074, China
`{fanyang,linguo,z_chen,wenbingtao}@hust.edu.cn`

## Abstract

Transformer architecture has shown great potential for many visual tasks, including point cloud registration. As an order-aware module, position encoding plays an important role in Transformer architecture applied to point cloud registration task. In this paper, we propose OIF-PCR, a one-inlier based position encoding method for point cloud registration network. Specifically, we first find one correspondence by a differentiable optimal transport layer, and use it to normalize each point for position encoding. It can eliminate the challenges brought by the different reference frames of two point clouds, and mitigate the feature ambiguity by learning the spatial consistency. Then, we propose a joint approach for establishing correspondence and position encoding, presenting an iterative optimization process. Finally, we design a progressive way for point cloud alignment and feature learning to gradually optimize the rigid transformation. The proposed position encoding is very efficient, requiring only a small addition of memory and computing overhead. Extensive experiments demonstrate the proposed method can achieve competitive performance with the state-of-the-art methods in both indoor and outdoor scenes.

## 1 Introduction

Point cloud registration is a fundamental research topic in many applications including scene reconstruction, autonomous driving, robotics, etc. The goal of it is to align two partially overlapped point clouds by estimating a rigid transformation between them. The commonly used feature-based point cloud registration involves two aspects: establishing point correspondences and estimating the rigid transformation. Although it has been widely studied in past decades, it is still challenging due to the difficulties such as low-overlap, repetitive patterns.

In recent years, learning-based works [28, 44, 50, 8] have made great advances in 3D representation. In addition to utilizing deep neural networks to learn better local feature descriptions [1, 9, 11], recent methods [18, 48, 29, 47] try to introduce Transformer [41] architecture to establish correspondences. One of the most important strategies of Transformer is position encoding, which has been proven crucial when applying Transformer to many computer vision tasks [22, 34, 35]. However, how to adopt the position encoding to the domain of 3D point cloud registration is still challenging. Different from other one-instance tasks (e.g., classification, segmentation and detection), which handle one image or point cloud in an inference, point cloud registration is a two-instance task. It needs to learn features of two point clouds simultaneously and match them. Since the two point clouds are under uncorrelated reference frames, the straightforward position encoding is not a good idea [24].

Recently, some methods attempt to design specific position encoding techniques [26, 24, 29] for point cloud registration. Geotransformer [29] calculates the relative positions between all points, then combines the position encoding with self-attention. This approach is equivalent to treating all points as reference points. It works well, but adds a lot of computational overhead due to the additional

---

[*]Corresponding author.

calculations about the distance and the angle similarity between points. DoPE [26] presents an efficient manner to learn a joint-origin as the reference point for position encoding. However, it depends on the centroid of point cloud to initialize the joint-origin, which is susceptible to noise in the low-overlapping scenes.

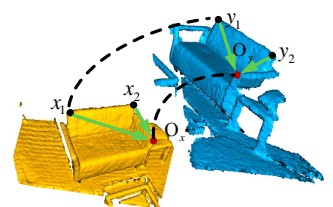

Figure 1: One-inlier based position learning illustration.

In this paper, we propose an efficient one-inlier based position encoding method for point cloud registration. We utilize a rigorously defined property of rigid transformation, i.e., it does not change the length of a vector (isometric isomorphism). As illustrated in Fig. 1, $(x_1, y_1)$ and $(O_x, O_y)$ are two correct correspondences, and the lengths of vector $\vec{x_1}$ $(x_1 \rightarrow O_x)$ and $\vec{y_1}$ $(y_1 \rightarrow O_y)$ are equal. Suppose $O_x$ and $O_y$ are selected as the reference points respectively, then we can find that $(x_1, y_1)$ is more likely to be a correct correspondence than $(x_2, y_1)$ only by position information without local feature, because the length of $\vec{x_1}$ is more close than $\vec{x_2}$ to that of $\vec{y_1}$. In this way, position information can benefit the establishing of correspondences when the local feature is ambiguous. This shows that only one inlier is enough to preserve the spatial consistency [2, 7] of correct correspondences and ensure the distinctiveness of point-wise features. Based on this observation, we only expect to find one inlier for position encoding. Specifically, we adopt the two points of this inlier to normalize each point cloud respectively, then use the normalized points for position encoding to enhance the feature of each point. The selected inlier plays two roles in the position encoding: 1) The two points in the inlier see the same spot of the same scenario, so positional differences caused by reference frames can be compensated based on their normalization. 2) It is feasible to do so to ensure spatial consistency, i.e., the relative spatial correlation between arbitrary point and the reference point is maintained across the two point clouds.

However, it is not easy to find an accurate inlier in advance as the reference point, which is a chicken-and-egg problem because determining the reference point depends on discriminative point-wise features. Thus, we propose a joint learning approach to optimize both position encoding and correspondences. It performs an iterative optimization process for finding good correspondences and position encoding. Specifically, we select multi-inliers by performing a differentiable optimal transport layer to produce a virtual correspondence instead of directly finding one correspondence, avoiding the unstable caused by the arbitrary selection. Then the two points corresponding to the correspondence are considered as the reference points. Next, the reference points are utilized to further encode point-wise geometric position features, and generate more discriminative features for point cloud. The updated point cloud features are further used for finding one virtual correspondence and performing position encoding again. Such circular optimization leads to more explicit position encoding. Furthermore, we use a progressive approach to gradually align the two point clouds (reference points), which reduces dependency on initial settings. During each step of the above optimization process, an estimated transformation is recovered from some correspondences to align the two point clouds. The position gap between reference points is narrowed in the process of alignment. We repeat the above step several times, which presents a progressive optimization process. Fig. 2 shows the overview of the proposed method.

Compared to the previous works on the position encoding for point cloud registration [26, 24, 29], this kind of strategy mainly has three advantages: 1) It is more reasonable for finding a correct correspondence than centroid estimation for position encoding, especially in the low-overlapping scenarios. 2) It is succinct and efficient, and competitive in terms of GPU memory usage and speed, which has been verified in the subsequent experiments. We only need to find an inlier to efficiently enhance the distinctiveness of features. 3) It is robust, which reduces the impact of outlier through constructing a virtual correspondence and avoids the dependency on the initialization through joint optimization.

In summary, our main contributions are as follows:

- An efficient position encoding with few correspondences for point cloud registration, which is light-weight and only relies on a few points for accurate position encoding.

- A joint optimization strategy that, optimizes the correspondence establishment and position encoding with point-wise feature encoding as the agent. By the way, the point cloud features are also continuously optimized.

- A progressive point cloud alignment approach, which progressively updates the point cloud positions so as to reduce dependency on initialization.

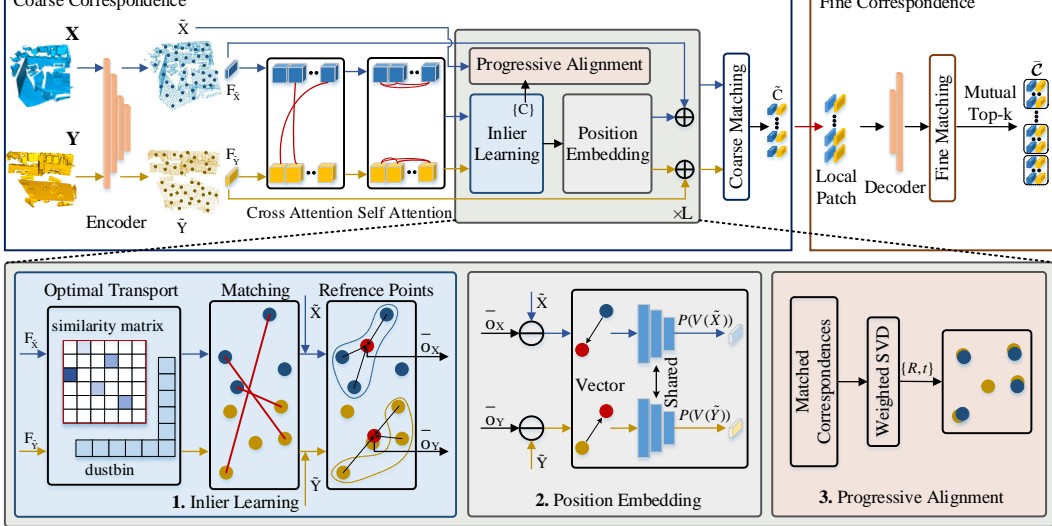

Figure 2: **Overview of the proposed method**. ($\oplus$: Feature reconstruction; $\ominus$: Point cloud normalization; {C}: Matched correspondences). The whole network is designed in a coarse-to-fine architecture for establishing correspondences by the proposed one-inlier based position encoding. The proposed position encoding mainly contains the following parts: **1.** An inlier learning approach is proposed to establish correspondences and output two reference points $\bar{\mathbf{o}}_{\mathbf{X}}$ and $\bar{\mathbf{o}}_{\mathbf{Y}}$ (Sec. 3.3). **2.** We utilize the predicted reference points $\bar{\mathbf{o}}_{\mathbf{X}}$ and $\bar{\mathbf{o}}_{\mathbf{Y}}$ to normalize the two point clouds for position embedding (Sec. 3.3). **3.** We leverage the established correspondences to iteratively align point clouds (reference points $\bar{\mathbf{o}}_{\mathbf{X}}$ and $\bar{\mathbf{o}}_{\mathbf{Y}}$) (Sec. 3.4). Finally, we repeat the above operations $L$ times through continual feature reconstruction, which presents a progressive joint optimization process (Sec. 3.5).

## 2 Related Work

**Local 3D feature descriptors.** Early traditional handcrafted descriptors widely summarize the local geometry in the form of histogram. Histograms usually contain statistical information such as the spatial distribution and geometric properties of points [16]. The descriptors represented with the 3D spatial distribution include Spin Image [19], 3D Shape Context [13], etc. Another category of descriptors includes Fast Point Feature Histograms (FPFHs) [30], SHOT [39], etc. Recent works attempt to encode 3D local descriptors in a data-driven way. Zeng et al. propose the pioneering work 3DMatch [49], which utilizes 3D Convolutional Neural Networks to encode local descriptors from local volumetric patches. PPFNet [11] introduces the rotation-invariant Point-Pair Features (PPF) [12] to encode local descriptors. Follow-up work PPF-FoldNet [10] incorporates FoldingNet [45] as backbone for unsupervised descriptor learning. 3DSN [15] leverages the voxelized Smoothed Density Value (SDV) to learn rotation-invariant local descriptors. FCGF [9] first utilizes the sparse 3D fully convolutional network to extract dense feature descriptors. SpinNet [1] proposes a 3D cylindrical convolution layer for learning rotation-invariant features.

**Learning-based point cloud registration.** Deep Closest Point [43] utilizes a graph neural network to extract point cloud features and a pointer network to predict correspondences. IDAM [23] proposes an iterative distance-aware similarity matrix convolution module to integrate the feature and Euclidean space into the point matching. Predator [18] leverages the attention mechanism to detect the overlapping region for sampling salient feature points. CoFiNet [48] utilizes a coarse-to-fine strategy for correspondence finding. Lepard [24] utilizes the Rotary positional encoding [34] to encode 3D positional information for feature learning. NgeNet [51] introduces a multi-scale architecture to extract features with multiple neighborhood sizes for adaptive selection of feature scale. REGTR [47] uses transformer to directly predict a set of point correspondences and estimates the rigid transformation using a variant of the Kabsch-Umeyama [20, 40] algorithm.

**Position embedding.** The idea of relative positional encodings first appeared in machine translation [32] and music generation [17]. They produce learned positional embeddings in the self-attention. DETR [6] applies fixed positional encodings [27, 4] to the input of the transformer architecture for object detection. Swin Transformer [25] adds relative position biases in similarity computation

of self-attention, improving performance over models without these biases. LoFTR [35] uses the 2D extension of the position encoding to produce position-dependent features for image matching. DoPE [26] proposes to calculate a joint-origin of all points for point cloud position encoding. Geotransformer [29] treats all points as reference points to calculate the relative positions between point pairs and combines them with self-attention to embed positional information into the features.

## 3 Approach

### 3.1 Problem Statement

Suppose there are two point clouds: source point cloud $\mathbf{X} = \{\mathbf{x}_i \in \mathbb{R}^3, i = 1, ..., M\}$ and target point cloud $\mathbf{Y} = \{\mathbf{y}_j \in \mathbb{R}^3, j = 1, ..., N\}$, where $M$ and $N$ are the point numbers of $\mathbf{X}$ and $\mathbf{Y}$, respectively. Our goal is to recover the rigid transformation $\mathbf{T} = \{\mathbf{R}, \mathbf{t}\}$ that best aligns the two point clouds, where $\mathbf{R} \in SO(3)$ represents the rotation component and $\mathbf{t} \in \mathbb{R}^3$ represents the translation component, as follows:

$$\arg\min_{\mathbf{R},\mathbf{t}} \sum_{(\bar{i},\bar{j}) \in \bar{\mathcal{C}}} \left\| \mathbf{y}_{\bar{j}} - (\mathbf{R} \cdot \mathbf{x}_{\bar{i}} + \mathbf{t}) \right\|_2, \tag{1}$$

where $\bar{\mathcal{C}}$ is a set of predicted correspondences between the source and target point clouds.

### 3.2 Pipeline

Fig. 2 illustrates our overview framework. Following [35, 48, 36], our method employs a coarse-to-fine manner to find correspondences. In the coarse stage, we first use the KPConv [38] to down-sample raw input points to uniformly distributed nodes $\tilde{\mathbf{X}} \in \mathbb{R}^{\tilde{M} \times 3}$ and $\tilde{\mathbf{Y}} \in \mathbb{R}^{\tilde{N} \times 3}$, and learn the associated features $\mathbf{F}_{\tilde{X}} \in \mathbb{R}^{\tilde{M} \times \tilde{D}}$ and $\mathbf{F}_{\tilde{Y}} \in \mathbb{R}^{\tilde{N} \times \tilde{D}}$. Then coarse correspondences are generated by matching the nodes. After obtaining the coarse correspondences from the previous stage, we refine them in fine level. Specifically, we assign each point in the fine level to its nearest nodes in geometry space. Each node will be expanded into a local patch by the point-to-node grouping strategy [21, 48], and the patch correspondences can be established by the matching of nodes. For each patch correspondence, we utilize an optimal transport layer [31] to calculate a soft assignment matrix between the fine-level points within this patch correspondence, then obtain the fine point correspondences by the mutual top-k selection. The final point correspondences are collected from all coarse correspondences.

Since the results of fine correspondences mainly depend on the coarse results, our method focuses on boosting the accuracy of the coarse results. Specifically, an inlier-based position learning approach is proposed to find an inlier and normalize the point clouds for position embedding (Sec. 3.3). Next, we leverage the matched correspondences (the byproduct of the "one-inlier" establishing) to progressively align point clouds (Sec. 3.4). After that, we propose to jointly optimize correspondence establishment and position embedding bridged by feature reconstruction (Sec. 3.5). Finally, we introduce the loss function for training the network (Sec. 3.6).

### 3.3 Inlier-based Position Learning

**Inlier learning.** As mentioned before, once a correspondence is an inlier, its corresponding points in the two point clouds can be considered as the reference points respectively. Since the two reference points are mapped from the same 3D point in different views, the matching points satisfied the ground-truth rigid transformation can preserve the spatial consistency (e.g., the spatial distance and orientation) relative to the respective reference points across the two point clouds. Thus, only one inlier is desired to compensate for the difference in 3D spatial position because the two point clouds in different reference systems can be normalized by the two points of the inlier. Given that the inlier is preemptively agnostic, our approach is mainly devoted to the mining of one correct correspondence. The cross-attention [41, 31] can encode the geometric consistency across the two point clouds by integrating the features of the other point cloud. We use it to enhance the distinctiveness of features. Formally, given the coarse features $\mathbf{F}_{\tilde{X}}$ and $\mathbf{F}_{\tilde{Y}}$, we use learnable matrices $\mathbf{W}_q$ to project the $\mathbf{F}_{\tilde{X}}$ to query $\mathbf{q}$, and use $(\mathbf{W}_k, \mathbf{W}_v)$ to project $\mathbf{F}_{\tilde{Y}}$ to key and value $(\mathbf{k}, \mathbf{v})$ as follows:

$$\mathbf{q}_i = \mathbf{F}_{\tilde{X}_i} \mathbf{W}_q, \mathbf{k}_j = \mathbf{F}_{\tilde{Y}_j} \mathbf{W}_k, \mathbf{v}_j = \mathbf{F}_{\tilde{Y}_j} \mathbf{W}_v, \tag{2}$$

where $\mathbf{W}_q, \mathbf{W}_k, \mathbf{W}_v \in \mathbb{R}^{\tilde{D} \times \tilde{D}}$. The coarse feature $\mathbf{F}_{\tilde{X}}$ is updated by $\mathbf{F}'_{\tilde{X}_i} = \mathbf{F}_{\tilde{X}_i} + \mathrm{MLP}(\sum_{j=1}^{\tilde{N}} \alpha_{ij} \mathbf{v}_j)$, where $\alpha_{ij} = \mathrm{softmax}(\mathbf{q}_i \mathbf{k}_j^T / \sqrt{\tilde{D}})$ is the feature similarity score between

$\mathbf{F}_{\tilde{X}_i}$ and $\mathbf{F}_{\tilde{Y}_j}$. $\mathrm{MLP}(\cdot)$ denotes a fully connected network. The cross-attention features for $\mathbf{F}_{\tilde{Y}}$ are computed and denoted in a similar way. Following the cross-attention we use a self-attention to aggregate the global context, which updates features in a similar way like cross-attention, but is performed inside the source and target point cloud independently.

After obtaining the strengthened features $\mathbf{F}'_{\tilde{X}}$ and $\mathbf{F}'_{\tilde{Y}}$, we leverage a differentiable optimal transport to find good correspondences. We compute the similarity matrix $\tilde{\mathbf{S}} \in \mathbb{R}^{\tilde{\mathbf{M}} \times \tilde{\mathbf{N}}}$ between $\mathbf{F}'_{\tilde{X}}$ and $\mathbf{F}'_{\tilde{Y}}$:

$$\tilde{\mathbf{S}}_{ij} = \mathbf{F}'_{\tilde{X}_i} \mathbf{F}'_{\tilde{Y}_j}{}^T, \tag{3}$$

then we append a new row and a new column as in [31], filled with a learnable dustbin parameter $\alpha$. The Sinkhorn algorithm [33] is adopted on $\tilde{\mathbf{S}}$ for searching the best solution of the optimal transportation problem. Finally, we obtain the soft matching scores $\bar{\mathbf{S}} \in \mathbb{R}^{\tilde{\mathbf{M}} \times \tilde{\mathbf{N}}}$ between the $\tilde{\mathbf{X}}$ and $\tilde{\mathbf{Y}}$ by dropping the last row and the last column of $\tilde{\mathbf{S}}$. Intuitively, the higher the matching score, the higher the probability that the associated correspondence is an inlier. But it is not always true due to the widely existing repetitive structures, textureless structures, etc. In order to enhance the robustness of the proposed algorithm and reduce the bias caused by the above issues, we propose to pick multiple reliable correspondences $\bar{\mathbf{C}}_{topk}$ with higher matching scores via top-k selection strategy. We use the selected correspondences to produce a virtual correspondence by an averaging operation of the point coordinates. The corresponding points of the virtual correspondence are regarded as the reference points $\bar{\mathbf{o}} = \{\bar{\mathbf{o}}_{\mathbf{X}}, \bar{\mathbf{o}}_{\mathbf{Y}}\}$, where the $\bar{\mathbf{o}}_{\mathbf{X}}, \bar{\mathbf{o}}_{\mathbf{Y}}$ denote the reference point in source and target point cloud, respectively.

**Position embedding.** After obtaining the reference points $\bar{\mathbf{o}}$, they are utilized to normalize the two point clouds respectively. Specifically, we use them to calculate point-wise position vector relative to the respective reference point for each node in the two point clouds:

$$V(\tilde{\mathbf{x}}_i) = \tilde{\mathbf{x}}_i - \bar{\mathbf{o}}_{\mathbf{X}}, V(\tilde{\mathbf{y}}_j) = \tilde{\mathbf{y}}_j - \bar{\mathbf{o}}_{\mathbf{Y}}. \tag{4}$$

Once the reference points and $(\tilde{\mathbf{x}}_i, \tilde{\mathbf{y}}_j)$ are inliers, i.e., they both meet the ground-truth rigid transformation between the point clouds, then the two vector $V(\tilde{\mathbf{x}}_i), V(\tilde{\mathbf{y}}_j)$ satisfy the isometric isomorphism property. It means that the isomorphism of $\tilde{\mathbf{x}}_i$ or $\tilde{\mathbf{y}}_j$ maintaining constant distance with respect to the respective reference point in different reference frames. Based on this property, the point-wise vectors normalized by the reference points are fed into position embedding layer for feature reconstruction. Specifically, we use an MLP with 4 hidden layers of feature size being 256 on each individual vector, and output an embedded geometric position feature. Formally, for vector $V(\tilde{\mathbf{x}}_i)$ and $V(\tilde{\mathbf{y}}_j)$, the corresponding feature is denoted as $\mathbf{P}(V(\tilde{\mathbf{x}}_i))$ and $\mathbf{P}(V(\tilde{\mathbf{y}}_j))$, separately. Then we use the embedded geometric position features to enhance the distinctiveness of the node features. In a simple and effective manner, we strengthen the node features by directly adding the geometric position features to the primal features, i.e., $\mathbf{H}_{\tilde{X}_i} = \mathbf{F}'_{\tilde{X}_i} + \mathbf{P}(V(\tilde{\mathbf{x}}_i)), \mathbf{H}_{\tilde{Y}_j} = \mathbf{F}'_{\tilde{Y}_j} + \mathbf{P}(V(\tilde{\mathbf{y}}_j))$.

### 3.4 Progressive Point Cloud Alignment

As mentioned before, the main challenge of position encoding is that the two point clouds are generally under irrelevant reference frames. We further solve this challenge by progressively aligning the point clouds in coordinate space. We expect to recover a rigid transformation with reliable correspondences $\bar{\mathbf{C}}_{topk}$ to make the position encoding easier.

**Position update.** After obtaining the coarse correspondences $\bar{\mathbf{C}}_{topk}$ with relatively higher matching scores, we use them to estimate the current optimal transformation $\bar{\mathbf{T}}_t$. Specifically, given the coarse correspondences $\bar{\mathbf{C}}_{topk}$ and corresponding matching scores, we use weighted SVD [5] to estimate the current optimal transformation $\bar{\mathbf{T}}_t$. Then we use the estimated $\bar{\mathbf{T}}_t = \{\bar{\mathbf{R}}_t, \bar{\mathbf{t}}_t\}$ to update the relative position between two point clouds:

$$\mathbf{x}_{i_t} = (\bar{\mathbf{R}}_t \cdot \mathbf{x}_i + \bar{\mathbf{t}}_t), \mathbf{x}_i \in \mathbf{X}, i = 1, ..., M. \tag{5}$$

Naturally, the reference points, i.e., the centroids $\bar{\mathbf{o}}$ of the selected patches in the two point clouds have also been updated:

$$\bar{\mathbf{o}}_{\mathbf{X}_t} = (\bar{\mathbf{R}}_t \cdot \bar{\mathbf{o}}_{\mathbf{X}} + \bar{\mathbf{t}}_t), \tag{6}$$

where $\bar{\mathbf{o}}_{\mathbf{X}_t}$ represents the transformed reference point of the source point cloud. As the two point clouds are roughly aligned by the estimated transformation, the position vectors of the two points in

each correct correspondence tend to be consistent. Thus, the encoded position feature can provide a more positive effect for establishing correspondences.

**Progressive alignment.** We repeat the above operation leveraging the enhanced features during the joint optimization (see Sec. 3.5) to progressively update the relative positions between the two point clouds. Specifically, every time before the position encoding operation, we perform an alignment between the two point clouds. We utilize the enhanced features to rebuild some reliable correspondences $\bar{\mathbf{C}}_{topk}$ and re-estimate the current optimal transformation $\bar{\mathbf{T}}_t$ for the alignment. Intuitively, the enhanced features are more distinctive than original features, so the latter estimated transformation is more accurate than that in previous stage. Thus, multiple transformations enable the two point clouds to be progressively aligned in coordinate space. In this way, the position encoding can also be optimized in the progressive alignment process.

### 3.5 Jointly Optimizing Position Encoding and Correspondences

Inlier learning and position encoding of the point clouds is a chicken-and-egg problem: the proposed position encoding for learning distinctive node features would require the knowledge about at least one correct correspondence; but meantime, establishing one accurate correspondence also relies on the correct position encoding of the point clouds for feature reconstruction. Either of them is hard to be solved in advance. In order to tackle this circular problem, we propose to jointly optimize the correspondences and position encoding by multiple feature rebuilding. As shown in Fig. 2, we incorporate the position encoding into the process of the node feature learning. We construct it together with inlier learning as a joint optimization task, with a view to improving the accuracy of coarse correspondences by enhancing the discrimination of position encoding. Specifically, the joint optimization is performed by enhancing the features of point clouds with 3D positional information. The updated more accurate vectors $V(\tilde{\mathbf{x}}_i), V(\tilde{\mathbf{y}}_j)$ with respect to the selected reference points facilitate better node feature encoding, which contributes to finding one more accurate correspondence. Meanwhile, the more accurate correspondence means better reference points $\bar{\mathbf{o}}$, which enables that the updated vectors $V(\tilde{\mathbf{x}}_i), V(\tilde{\mathbf{y}}_j)$ gradually satisfy the isometric isomorphism property. It will result in more discriminative node-wise features. During the above process, the features of the point clouds are continuously optimized.

### 3.6 Loss Functions

The training loss function $\mathcal{L} = \mathcal{L}_c + \mathcal{L}_f$ is composed of the coarse correspondence loss $\mathcal{L}_c$ and fine correspondence loss $\mathcal{L}_f$.

**Coarse correspondence loss.** To supervise the patch-wise feature descriptors for coarse correspondences, we follow [18, 29] and use a variant of the circle loss [37]. Considering again an point cloud pair $\mathbf{X}$ and $\mathbf{Y}$, we first pick the patch $\mathbf{P}_i^{\mathbf{X}}$ (denotes the patch corresponding to the node $\tilde{\mathbf{x}}_i \in \tilde{\mathbf{X}}$) that has at least one positive patch in $\mathbf{Y}$ to form a collection of patches $\mathcal{P}$. For each patch $\mathbf{P}_i^{\mathbf{X}} \in \mathcal{P}$, the set of its positive (share at least 10% overlap) and negative (do not share overlap) patches in $\mathbf{Y}$ are denoted as $\varepsilon_p^i$ and $\varepsilon_n^i$, respectively. The loss of coarse correspondence on $\mathbf{X}$ is then calculated as:

$$\mathcal{L}_c^{\mathbf{X}} = \frac{1}{|\mathcal{P}|} \sum_{i=1}^{|\mathcal{P}|} \log[1 + \sum_{\mathbf{P}_j^{\mathbf{Y}} \in \varepsilon_p^i} e^{\lambda_i^j \beta_p^{i,j}(d_i^j - \Delta_p)} \cdot \sum_{\mathbf{P}_k^{\mathbf{Y}} \in \varepsilon_n^i} e^{\beta_n^{i,k}(\Delta_n - d_i^k)}], \tag{7}$$

where $d_i^j = \left\| \mathbf{H}_{\tilde{X}_i} - \mathbf{H}_{\tilde{Y}_j} \right\|_2$ represents the distance in feature space, and $\lambda_i^j = (o_i^j)^{\frac{1}{2}}$, and $o_i^j$ denotes the overlap ratio between $\mathbf{P}_i^{\mathbf{X}}$ and $\mathbf{P}_j^{\mathbf{Y}}$. $\Delta_p, \Delta_n$ are positive and negative margins, respectively. The weights for each positive and negative sample are calculated individually: $\beta_p^{i,j} = \gamma(d_i^j - \Delta_p)$ and $\beta_n^{i,k} = \gamma(\Delta_n - d_i^k)$, with the empirical margins $\Delta_p = 0.1, \Delta_n = 1.4$. $\gamma$ is a hyper-parameter. The loss $\mathcal{L}_c^{\mathbf{Y}}$ on $\mathbf{Y}$ is computed in the same way. The total loss is $\mathcal{L}_c = (\mathcal{L}_c^{\mathbf{X}} + \mathcal{L}_c^{\mathbf{Y}})/2$.

**Fine correspondence loss.** For fine correspondence, we first determine the ground-truth coarse correspondences using the ground-truth relative transformations. Then we extract a set of ground-truth fine correspondences $\mathcal{M}_i$ in each correspondence patches. We label the unmatched points in this two patches as $\mathcal{I}_i$ and $\mathcal{J}_i$. The fine correspondence loss is computed as:

$$\mathcal{L}_f = -\frac{1}{C} \sum_{i=1}^{C} (\sum_{(x,y) \in \mathcal{M}_i} \log \bar{\mathbf{P}}_{x,y}^i + \sum_{x \in \mathcal{I}_i} \log \bar{\mathbf{P}}_{x,n_i+1}^i + \sum_{y \in \mathcal{J}_i} \log \bar{\mathbf{P}}_{m_i+1,y}^i), \tag{8}$$

where $C$ is the number of the patch pairs. $\bar{\mathbf{P}}^i \in \mathbb{R}^{(m_i+1)\times(n_i+1)}$ is the soft assignment matrix of the $i$-th patch correspondence. $m_i$ and $n_i$ represent the number of the fine level points in this patch pair.

Table 1: Evaluation results on 3DMatch and 3DLoMatch datasets.

| | | 3DMatch | | | | | | | | | | | | | | |
|---|---|---|---|---|---|---|---|---|---|---|---|---|---|---|---|---|
| | | RR (%) ↑ | | | | | FMR (%) ↑ | | | | | IR (%) ↑ | | | | |
| | # Samples | 5000 | 2500 | 1000 | 500 | 250 | 5000 | 2500 | 1000 | 500 | 250 | 5000 | 2500 | 1000 | 500 | 250 |
| descriptor | 3DSN [15] | 78.4 | 76.2 | 71.4 | 67.6 | 50.8 | 95.0 | 94.3 | 92.9 | 90.1 | 82.9 | 36.0 | 32.5 | 26.4 | 21.5 | 16.4 |
| | FCGF [9] | 85.1 | 84.7 | 83.3 | 81.6 | 71.4 | 97.4 | 97.3 | 97.0 | 96.7 | 96.6 | 56.8 | 54.1 | 48.7 | 42.5 | 34.1 |
| | D3Feat [3] | 81.6 | 84.5 | 83.4 | 82.4 | 77.9 | 95.6 | 95.4 | 94.5 | 94.1 | 93.1 | 39.0 | 38.8 | 40.4 | 41.5 | 41.8 |
| | SpinNet [1] | 88.6 | 86.6 | 85.5 | 83.5 | 70.2 | 97.6 | 97.2 | 96.8 | 95.5 | 94.3 | 47.5 | 44.7 | 39.4 | 33.9 | 27.6 |
| | YOHO [42] | 90.8 | 90.3 | 89.1 | 88.6 | 84.5 | **98.2** | 97.6 | 97.5 | 97.7 | 96.0 | 64.4 | 60.7 | 55.7 | 46.4 | 41.2 |
| end-to-end | REGTR [47] | | | 92.0 | | | | | - | | | | | - | | |
| | Predator [18] | 89.0 | 89.9 | 90.6 | 88.5 | 86.6 | 96.6 | 96.6 | 96.5 | 96.3 | 96.5 | 58.0 | 58.4 | 57.1 | 54.1 | 49.3 |
| | CoFiNet [48] | 89.3 | 88.9 | 88.4 | 87.4 | 87.0 | 98.1 | **98.3** | **98.1** | 98.2 | 98.3 | 49.8 | 51.2 | 51.9 | 52.2 | 52.2 |
| | GeoTrans [29] | 92.0 | 91.8 | **91.8** | 91.4 | **91.2** | 97.9 | 97.9 | 97.9 | 97.9 | 97.6 | **71.9** | **75.2** | **76.0** | **82.2** | **85.1** |
| | Ours | **92.4** | **91.9** | **91.8** | **92.1** | **91.2** | 98.1 | 98.1 | 97.9 | **98.4** | **98.4** | 62.3 | 65.2 | 66.8 | 67.1 | 67.5 |
| | | 3DLoMatch | | | | | | | | | | | | | | |
| | | RR (%) ↑ | | | | | FMR (%) ↑ | | | | | IR (%) ↑ | | | | |
| | # Samples | 5000 | 2500 | 1000 | 500 | 250 | 5000 | 2500 | 1000 | 500 | 250 | 5000 | 2500 | 1000 | 500 | 250 |
| descriptor | 3DSN [15] | 33.0 | 29.0 | 23.3 | 17.0 | 11.0 | 63.6 | 61.7 | 53.6 | 45.2 | 34.2 | 11.4 | 10.1 | 8.0 | 6.4 | 4.8 |
| | FCGF [9] | 40.1 | 41.7 | 38.2 | 35.4 | 26.8 | 76.6 | 75.4 | 74.2 | 71.7 | 67.3 | 21.4 | 20.0 | 17.2 | 14.8 | 11.6 |
| | D3Feat [3] | 37.2 | 42.7 | 46.9 | 43.8 | 39.1 | 67.3 | 66.7 | 67.0 | 66.7 | 66.5 | 13.2 | 13.1 | 14.0 | 14.6 | 15.0 |
| | SpinNet [1] | 59.8 | 54.9 | 48.3 | 39.8 | 26.8 | 75.3 | 74.9 | 72.5 | 70.0 | 63.6 | 20.5 | 19.0 | 16.3 | 13.8 | 11.1 |
| | YOHO [42] | 65.2 | 65.5 | 63.2 | 56.5 | 48.0 | 79.4 | 78.1 | 76.3 | 73.8 | 69.1 | 25.9 | 23.3 | 22.6 | 18.2 | 15.0 |
| end-to-end | REGTR [47] | | | 64.8 | | | | | - | | | | | - | | |
| | Predator [18] | 59.8 | 61.2 | 62.4 | 60.8 | 58.1 | 78.6 | 77.4 | 76.3 | 75.7 | 75.3 | 26.7 | 28.1 | 28.3 | 27.5 | 25.8 |
| | CoFiNet [48] | 67.5 | 66.2 | 64.2 | 63.1 | 61.0 | 83.1 | 83.5 | 83.3 | 83.1 | 82.6 | 24.4 | 25.9 | 26.7 | 26.8 | 26.9 |
| | GeoTrans [29] | 75.0 | 74.8 | 74.2 | 74.1 | 73.5 | **88.3** | **88.6** | **88.8** | **88.6** | **88.3** | **43.5** | **45.3** | **46.2** | **52.9** | **57.7** |
| | Ours | **76.1** | **75.4** | **75.1** | **74.4** | **73.6** | 84.6 | 85.2 | 85.5 | 86.6 | 87.0 | 27.5 | 30.0 | 31.2 | 32.6 | 33.1 |

## 4 Experiments

We evaluate our approach on indoor 3DMatch [49] and 3DLoMatch [18] benchmarks (Sec. 4.1), and outdoor KITTI odometry [14] benchmark (Sec. 4.2). The point cloud pairs in 3DMatch have > 30% overlap, while those in 3DLoMatch have low overlap of $10\% \sim 30\%$. KITTI odometry is an outdoor driving scenario sparse point cloud dataset acquired by LiDAR. Details about the datasets and implementation are provided in the supplementary material.

### 4.1 3DMatch and 3DLoMatch

**Metrics.** Our main metric, related to the practical purpose of point cloud registration, is *Registration Recall* (RR), i.e., the percentage of point cloud pairs whose transformation error (RMSE) is below 0.2 m. It is the most important metric in point cloud registration task [18, 48], because it directly measures the final registration performance. Following [3, 18], we also report *Feature Matching Recall* (FMR), i.e., the fraction of point cloud pairs whose *Inlier Ratio* is larger than 5%, and *Inlier Ratio* (IR), defined as the fraction of correct correspondences (residuals are less than 0.1 m under the ground truth transformation) among the putative matches.

**Comparisons to the state-of-the-art approaches.** We compare the registration results of our method with recent state-of-the-art approaches including: 3DSN [15], FCGF [9], D3Feat [3], SpinNet [1], YOHO [42], Predator [18][2], CoFiNet [48], REGTR [47] and GeoTransformer (GeoTrans in the table) [29] in Tab. 1. We report the results under different numbers of sampled correspondences. Since REGTR [47] needs to take all the final downsampled points for transformation estimation instead of sampling, we only provide the results in the original paper. Following [3, 18], we run RANSAC-50k to estimate the rigid transformation. When compared with local 3D descriptors, our method outperforms all the approaches on both datasets. For end-to-end methods, the proposed method and

---

[2]The Predator had a bug in its initial version, we provides the updated results in its corrected version.

GeoTransformer [29] outperform other methods by a large margin on all the metrics, especially in the low-overlap scenarios (3DLoMatch). This is because our method and GeoTransformer both consider the position information in the network, which demonstrates the importance of position encoding for point cloud registration.

It is worth noting that our method performs worse than GeoTransformer [29] on Inlier Ratio (IR). However, when referring to the most important metric Registration Recall (RR) which better reflects the final performance on point cloud registration, the proposed method achieves the state-of-the-art performance. It can be explained by the fact that the inlier rate is not total positively-correlated with the registration recall. As pointed out in [18, 48, 29], besides the inlier rate, the other important issue for registration is the sparsity of the correspondences. Aggregated inliers may improve the inlier rate, but are not necessarily beneficial for the registration. Since our position encoding can introduce spatial consistency, it is possible to find matching pairs that cannot be found by local features alone. Thus, although the inlier rate of our method is not the highest, our method still achieves the best performance in terms of Registration Recall.

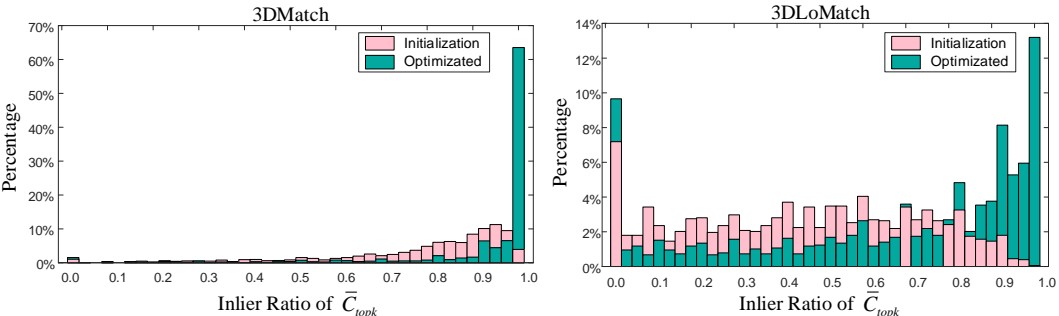

Figure 3: The scene distribution in different inlier ratios of $\bar{\mathbf{C}}_{topk}$ on 3DMatch and 3DLoMatch.

**Initialization and optimization.** Our method performs an iterative optimization for finding reference points and reconstructing features. In order to clearly show the effectiveness of the optimization process, we count the scene frequency of different inlier ratios in the initial and optimized $\bar{\mathbf{C}}_{topk}$ (described in Sec. 3.3) and provide the histograms of scene distributions in Fig. 3. Specifically, we compute the inlier ratio of the initial and optimized $\bar{\mathbf{C}}_{topk}$ respectively for each scene on 3DMatch and 3DLoMatch datasets. Then we count the number of scenes with different inlier ratios of $\bar{\mathbf{C}}_{topk}$ on both datasets and present them in percentage form. In the initial stage, we only use the features learned by the backbone to find $\bar{\mathbf{C}}_{topk}$. On 3DMatch dataset, the inlier ratio is between 0.7 and 1.0 in most scenarios, and that of 3DLoMatch dataset is about 0 to 0.8. This shows that the initial backbone features in most scenes are enough to provide acceptable initial reference points. After the joint optimization, the node-wise features are reconstructed with the normalized 3D positional information. The inlier ratio is increased to $0.9 \sim 1.0$ on the 3DMatch dataset, and the inlier ratio in most scenes of 3DLoMatch is raised to $0.6 \sim 1.0$. It demonstrates that the proposed optimization method can significantly improve the matching accuracy of the reference points. Furthermore, we visualize the iterative alignment of the reference points (red and green patches) in Fig. 4. This indicates that the alignment error between two reference points gradually decreases during the optimization process.

**Computing overhead and runtime.** We evaluate the parameters, GPU memory usage and runtime of our method. Tab. 2 shows the comparison results with other methods. Considering the practical application, we provide the peck of GPU memory usage. On parameters, GeoTransformer [29] is 27% more than ours, but our method still achieves a competitive registration accuracy compared to it and significantly out-

Table 2: Evaluation of parameters [M], GPU memory [GB] and runtime [s] on 3DMatch and 3DLoMatch.

| Methods | Params | Mem. | Time | | |
| --- | --- | --- | --- | --- | --- |
| | | | Model | Pose | Total |
| D3Feat [3] | 27.3 | 7.26 | 0.0449 | 1.832 | 1.877 |
| Predator [18] | 7.43 | 7.78 | 0.0468 | 2.666 | 2.713 |
| CoFiNet [48] | 5.48 | 3.72 | 0.218 | 0.705 | 0.923 |
| GeoTransformer [29] | 9.83 | 12.73 | 0.247 | 1.743 | 1.99 |
| Ours | 7.75 | 4.85 | 0.141 | 0.414 | 0.555 |

performs other methods. On GPU memory usage, our method ranks 2nd only behind CoFiNet [48], and is only about 40% of GeoTransformer's [29]. With such low memory usage, our method still achieves the state-of-the-art registration performance. On runtime, our method achieves the fastest registration speed. This demonstrates the superiority of our method in both accuracy and speed.

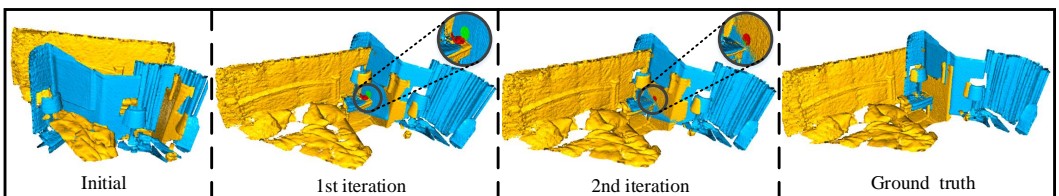

| Initial | 1st iteration | 2nd iteration | Ground truth |

Figure 4: Qualitative process of the iterative alignment.

## 4.2 KITTI

**Metrics.** Following [18], we evaluate our method with three metrics: 1) *Relative Rotation Error* (RRE), the geodesic distance between estimated and ground-truth rotation matrices, 2) *Relative Translation Error* (RTE), the Euclidean distance between estimated and ground-truth translation vectors, and 3) *Registration Recall* (RR), which has mentioned before, the thresholds are set as RRE < 5° and RTE < 2m.

**Registration results.** In Tab. 3, we compare our method with state-of-the-art approaches, including: 3DFeat-Net [46], FCGF [9], D3Feat [3], SpinNet [1], Predator [18], CoFiNet [48] and GeoTransformer [29]. Our method performs the best for all metrics, which proves that our method is scene-agnostic and maintains good registration accuracy in strongly differing scenarios.

Table 3: Evaluation results on KITTI odometry.

| Methods | RTE(cm) ↓ | RRE (°) ↓ | RR(%) ↑ |
|---|---|---|---|
| 3DFeat-Net [46] | 25.9 | 0.25 | 96.0 |
| FCGF [9] | 9.5 | 0.30 | 96.6 |
| D3Feat [3] | 7.2 | 0.30 | **99.8** |
| SpinNet [1] | 9.9 | 0.47 | 99.1 |
| Predator [18] | 6.8 | 0.27 | **99.8** |
| CoFiNet [48] | 8.5 | 0.41 | **99.8** |
| GeoTransformer [29] | 6.8 | 0.24 | **99.8** |
| Ours | **6.5** | **0.23** | **99.8** |

## 4.3 Ablation study

Table 4 shows the results of the ablation studies on the 3DMatch and 3DLoMatch datasets. We remove one component at each time from the complete pipeline (Full) to measure the corresponding contribution. Removing the progressive alignment leads to lower Registration Recall on both datasets, which means that aligning reference points in 3D space is beneficial for more accurate positional embedding. Without the joint optimization, the error of the estimated rigid transformation drastically increases and the matching inlier ratio decreases significantly. The joint optimization plays an important role in learning discriminative features. It avoids a lot of incorrect matches arising from 3D structural ambiguity. Finally, we replace the proposed one-inlier based position encoding with the centroid based position encoding, i.e. using the centroids as reference points instead of learning an inlier. Since the centroids do not constitute a correct correspondence for a pair of partially overlapping point clouds, this alternative results in worse performance.

Table 4: Ablation study for each component, tested with # Samples = 5000.

| | 3DMatch | | | 3DLoMatch | | |
|---|---|---|---|---|---|---|
| Methods | RR(%) ↑ | FMR(%) ↑ | IR(%) ↑ | RR(%) ↑ | FMR(%) ↑ | IR(%) ↑ |
| Full | 92.4 | 98.1 | 62.3 | 76.1 | 84.6 | 27.5 |
| w/o progressive alignment | 91.4 | 98.2 | 61.8 | 74.2 | 84.1 | 27.1 |
| w/o joint optimization | 91.1 | 97.5 | 58.9 | 72.6 | 83.6 | 25.7 |
| w/o associated reference points | 89.8 | 97.2 | 57.1 | 69.8 | 82.7 | 24.2 |

## 4.4 Relationship between the registration and correspondences

In order to analyze the relationship between the registration results and the established correspondences, we count the scene frequency of different inlier ratios for Geotransformer and our method, and present the curve charts of scene distributions in Fig. 5. Specifically, we count the number of scenes with different inlier ratios on 3DMatch and 3DLoMatch datasets, and present them in frequency form. Besides, we also calculate the scene registration recall of different inlier ratios and provide the curve charts in Fig. 6. On 3DMatch dataset, the inlier ratio of our method is between 0.2 and 0.9 in most scenarios, while the inlier ratio of Geotransformer is about 0.5 to 1.0. On 3DLoMatch, the inlier ratio of most scenarios for our method is between 0.05 and 0.6. For Geotransformer, it has more scenarios where the inlier ratio is 0.5 ~ 0.9. In fact, the inlier ratio is counted by averaging all the point cloud pairs. Geotransformer [29] has higher inlier ratio in some scenarios, which causes the mean inlier ratio to be higher than ours. However, higher inlier ratio is not a necessary condition for high registration recall. As show in the Fig. 6, we can observe that our method can achieve better

registration recall than Geotransformer with the same inlier ratio (especially the interval [0.15, 0.5] on 3DLoMatch dataset), which shows that the inliers of our method may be better distributed for model estimation.

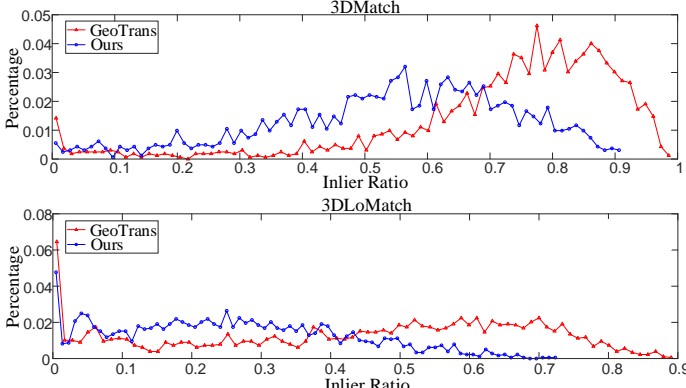

Figure 5: The scene distribution of different inlier ratios on 3DMatch and 3DLoMatch datasets.

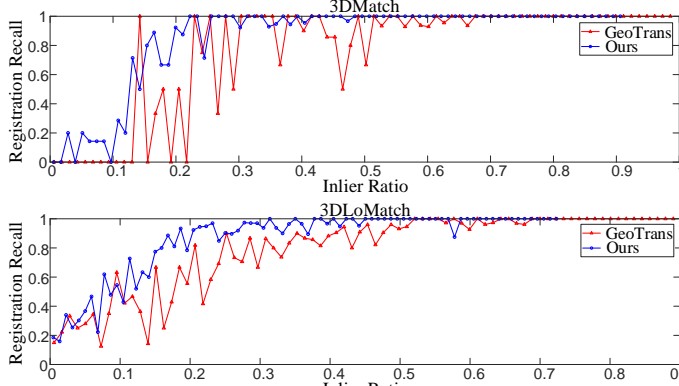

Figure 6: The scene registration recall of different inlier ratios on 3DMatch and 3DLoMatch datasets.

## 5 Conclusion

In this paper, we have introduced a simple but efficient position encoding for point cloud registration. In contrast to previous methods that rely on a lot of computing resources or need to meet stringent conditions, our method is light-weight and easy to implement. Our method only relies on a few correspondences: we show that one inlier is enough to realize accurate position encoding for learning distinguishable features. We utilize a differentiable optimal transport layer to find a good correspondence, which normalizes each point for position encoding. It builds a reliable bridge between point clouds in different reference frames, and preserves spatial consistency to reduce feature ambiguity. Besides, we propose a joint optimization between correspondence establishment and position encoding, and along with that, the point cloud features are also iteratively optimized. Furthermore, a progressive alignment approach is proposed to update point cloud positions and features alternately. The proposed method achieves a competitive registration performance with more complex state-of-the-art pipelines on indoor and outdoor datasets.

**Limitations.** Our method adopts a coarse-to-fine manner to find correspondences. However, the final point correspondences are largely subject to the coarse correspondences. Once a patch correspondence is incorrect, it would result in incorrect point correspondences. These incorrect point correspondences would have a potential negative impact on the final registration accuracy. Besides, our method would fail in the case of the overlapping region between two point clouds is too small or the overlapping region is ambiguous structure. A possible solution is to expand the overlapping region via point cloud completion.

## 6 Acknowledgment

This work was in part supported by the National Natural Science Foundation of China under Grants 62176096.

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
