# One-Inlier is First: Towards Efficient Position Encoding for Point Cloud Registration —Supplementary Material—

**Fan Yang**   **Lin Guo**   **Zhi Chen**   **Wenbing Tao**[*]
School of Artificial Intelligence and Automation
Huazhong University of Science and Technology, Wuhan 430074, China
{fanyang,linguo,z_chen,wenbingtao}@hust.edu.cn

## A   Appendix

In this supplementary material, we first provide the rigorous definitions of evaluation metrics (Sec. A.1), then describe the network architectures (Sec. A.2), implementation details (Sec. A.3) and datasets (Sec. A.4) in detail. We further provide additional ablation studies (Sec. A.5) and experimental results (Sec. A.6). We then discuss the broader impact (Sec. A.7) of our work. Finally, we show more qualitative results of registration on 3DMatch, 3DLoMatch and KITTI (Sec. A.8).

### A.1   Evaluation Metrics

Following [1, 3, 9], we evaluate the proposed method with different metrics on 3DMatch/3DLoMatch and KITTI. On 3DMatch and 3DLoMatch, we report *Inlier Ratio* (IR), *Feature Matching Recall* (FMR) and *Registration Recall* (RR). On KITTI, the *Relative Rotation Error* (RRE), *Relative Translation Error* (RTE) and *Registration Recall* (RR) are reported.

**3DMatch and 3DLoMatch**   *Inlier Ratio* (IR) measures the fraction of correct correspondences in the putative correspondences. A correspondence $(\mathbf{x}_i, \mathbf{y}_j) \in \mathcal{C}$ is considered correct if the distance between the two points is smaller than $\tau_1 = 10$cm under the ground truth transformation $\mathbf{T}^* = \{\mathbf{R}^*, \mathbf{t}^*\}$ between $\mathbf{X}$ and $\mathbf{Y}$. Given the predicted correspondence set $\mathcal{C}$, *Inlier Ratio* of the point cloud pair $(\mathbf{X}, \mathbf{Y})$ can be calculated by:

$$\text{IR} = \frac{1}{|\mathcal{C}|} \sum_{(\mathbf{x}_i, \mathbf{y}_j) \in \mathcal{C}} [\|\mathbf{T}^*(\mathbf{x}_i) - \mathbf{y}_j\|_2 < \tau_1], \tag{1}$$

where $[\cdot]$ is the Iversion bracket and $\|\cdot\|_2$ is the Euclidean norm.

*Feature Matching Recall* (FMR) measures the fraction of point cloud pairs whose *Inlier Ratio* is above a certain threshold $\tau_2 = 5\%$. It indicates the likelihood that the optimal transformation between two point clouds can be recovered using some robust estimator such as RANSAC [6], based on the predicted correspondence set $\mathcal{C}$:

$$\text{FMR} = \frac{1}{\text{N}} \sum_{i=1}^{\text{N}} [\text{IR}_i > \tau_2], \tag{2}$$

where N is the number of all point cloud pairs.

*Registration Recall* (RR) is the most important and reliable metric, because it directly evaluates the quality of the actual task of point cloud registration. *Registration Recall* measures the fraction of

---

[*]Corresponding author.

36th Conference on Neural Information Processing Systems (NeurIPS 2022).

correctly registered point cloud pairs. Two point clouds are considered as correctly registered if their transformation error is below $\tau_3 = 0.2$m. The transformation error is calculated as the Root Mean Square Error (RMSE) of the ground truth correspondences $\mathcal{C}^*$ after applying the estimated transformation $\mathbf{T}$:

$$\text{RMSE} = \sqrt{\frac{1}{|\mathcal{C}^*|} \sum_{(\mathbf{x}_i^*, \mathbf{y}_j^*) \in \mathcal{C}^*} \left\| \mathbf{T}(\mathbf{x}_i^*) - \mathbf{y}_j^* \right\|_2^2}, \tag{3}$$

$$\text{RR} = \frac{1}{\text{N}} \sum_{i=1}^{\text{N}} [\text{RMSE}_i < \tau_3], \tag{4}$$

Besides, we keep with the original evaluation protocol in 3DMatch [26], and exclude the immediately adjacent point clouds since they have very high overlap ratios.

**KITTI**    *Relative Rotation Error* (RRE) measures the geodesic distance in degrees between the estimated and ground truth rotation matrices:

$$\text{RRE} = \arccos \left( \frac{\text{trace}(\mathbf{R}^T \mathbf{R}^*) - 1}{2} \right), \tag{5}$$

where $\mathbf{R}$ denotes the estimated rotation matrix.

*Relative Translation Error* (RTE) measures the Euclidean distance between the estimated and ground truth translation vectors:

$$\text{RTE} = \left\| \mathbf{t} - \mathbf{t}^* \right\|_2, \tag{6}$$

where $\mathbf{t}$ denotes the estimated translation vector.

*Registration Recall* (RR) on KITTI measures the fraction of correctly registered point cloud pairs whose RRE and RTE are both below certain thresholds:

$$\text{RR} = \frac{1}{\text{N}} \sum_{i=1}^{\text{N}} [\text{RRE}_i < 5° \wedge \text{RTE}_i < 2\text{m}]. \tag{7}$$

Following [1, 3, 9, 12, 25, 14], we compute the mean RRE and RTE only with the correctly registered point cloud pairs on KITTI.

## A.2   Network Architecture Details

The detailed network architecture is depicted in Fig. 5. We use an encoder-decoder architecture based on KPConv backbone [20] for feature extraction. The voxel-grid subsampling [20] is used to downsample the point clouds. As a preprocessing step, the input point clouds are first downsampled with a voxel size of 2.5cm on 3DMatch/3DLoMatch and 30cm on KITTI before being fed into the backbone. We use the same voxel size and convolution radius in the KPConv backbone as in [9], i.e., the voxel size is doubled in each downsampling operation. Therefore each point downsampling results in the same outcome as in [9]. The upsampling in the decoder is performed by querying the corresponding feature of the nearest point from the previous layer. We follow the configurations of KPConv in [9] and apply group normalization [23] after each KPConv layer. Since our network does not rely on a lot of self- and cross- attention modules for feature aggregation and only needs to find an inlier for feature reconstruction, our approach is lightweight and has a fast inference speed.

**Coarse Matching**    After $L$ times feature reconstruction, we leverage the reconstructed features to find coarse correspondences. The score matrix $\mathbf{S} \in \mathbb{R}^{\tilde{M} \times \tilde{N}}$ between the reconstructed features $\mathbf{H}_{\tilde{X}}$ and $\mathbf{H}_{\tilde{Y}}$ is first calculated by $\mathbf{S}_{ij} = \exp(- \left\| \mathbf{H}_{\tilde{X}_i} - \mathbf{H}_{\tilde{Y}_j} \right\|_2^2)$. Then we apply a dual-softmax operator[15, 19], i.e., applying softmax on both dimensions to obtain the probability of soft mutual nearest neighbor matching. Formally, the probability of coarse matching $\boldsymbol{P}$ is computed as:

$$\boldsymbol{P}_{ij} = \text{softmax}(\mathbf{S}_{i \cdot})_j \cdot \text{softmax}(\mathbf{S}_{\cdot j})_i. \tag{8}$$

Based on the matching probability $\boldsymbol{P}$, we select coarse matches corresponding to the top-k largest entries in $\boldsymbol{P}$:

$$\tilde{\mathcal{C}} = \{(\tilde{\mathbf{x}}_i, \tilde{\mathbf{y}}_j) | (i, j) \in \text{topk}_{i,j}(\boldsymbol{P}_{ij})\}. \tag{9}$$

**Fine Matching**  After establishing coarse correspondences, these correspondences are refined to point level. Those refined correspondences are then used for point cloud registration. For each coarse correspondence $(\tilde{\mathbf{x}}_i, \tilde{\mathbf{y}}_j)$, its refined point correspondences are extracted from the corresponding patches $\mathbf{P}_i^{\mathbf{X}}$ and $\mathbf{P}_j^{\mathbf{Y}}$. We first compute a similarity matrix $\mathbf{s} \in \mathbb{R}^{m_i \times n_j}$ using the feature matrices $\mathbf{F}_i^{\mathbf{X}} \in \mathbb{R}^{m_i \times d}$ and $\mathbf{F}_j^{\mathbf{Y}} \in \mathbb{R}^{n_j \times d}$ of the two patches:

$$\mathbf{s} = \mathbf{F}_i^{\mathbf{X}}(\mathbf{F}_j^{\mathbf{Y}})^T/\sqrt{d}, \tag{10}$$

where $m_i = |\mathbf{P}_i^{\mathbf{X}}|$, $n_j = |\mathbf{P}_j^{\mathbf{Y}}|$. $d$ is the feature dimension.

Then we augment the similarity matrix $\mathbf{s}$ by appending a new row and new column as [16], filled with a learnable dustbin parameter $\alpha$. The fine point matching problem can be formulated as an optimal transport problem. We run the Sinkhorn algorithm [18] on $\mathbf{s}$ to solve this problem. Then the final soft matching score $\bar{\mathbf{s}} \in \mathbb{R}^{m_i \times n_j}$ is obtained by dropping the last row and the last column of $\mathbf{s}$. We utilize $\bar{\mathbf{s}}$ as the confidence matrix of the candidate matches and extract the fine point correspondences by the mutual top-k selection strategy. A point correspondence is selected if the corresponding matching score is among the $k$ largest entries of both the row and the column. The final point correspondence set $\bar{\mathcal{C}}$ is a collection of all the refined correspondences from each coarse correspondence.

## A.3  Implementation Details

We implement the proposed network in PyTorch [13]. All experiments are conducted on an Intel(R) Xeon(R) Platinum 8255C CPU and an NVIDIA RTX 2080Ti GPU. We train our network using Adam optimizer [10] with 40 epochs on 3DMatch/3DLoMatch and 80 epochs on KITTI. The batch size is 1 and the weight decay is $10^{-6}$. The initial learning rate is $10^{-4}$ and exponentially decayed by 0.05 after each epoch on 3DMatch/3DLoMatch and every 4 epochs on KITTI. We apply training data augmentation as in [9]. We use the matching radius $\tau = 5$cm for 3DMatch/3DLoMatch and $\tau = 60$cm for KITTI to determine overlapping during the generation of both coarse-level and fine-level ground truth matches. We randomly sample 128 coarse correspondences with patch size being 64 during training. we sample point correspondences with probability proportional to the matching score during testing. The proposed joint optimization is repeated $L = 2$ times. The number of correspondences in $\bar{\mathbf{C}}_{topk}$ is 50. We run 100 iterations of the Sinkhorn Algorithm.

## A.4  Datasets

**3DMatch and 3DLoMatch**  3DMatch [26] is a collection of 62 scenes from SUN3D [24], 7-Scenes [17], RGB-D Scenes v.2 [11], Analysis-by-Synthesis [21], BundleFusion [5] and Halbel et al. [8] (Tab. 1). Individual scenes are captured from diverse indoor spaces (e.g. kitchens, offices, bedrooms, living rooms) and different sensors (e.g. Microsoft Kinect, Structure Sensor, Asus Xtion Pro Live, Intel Realsense). Each scene in 3DMatch is split into point cloud fragments, which are generated by fusing 50 consecutive depth frames using TSDF volumetric fusion [4]. We utilize the voxel-grid downsampled point clouds from [9] for training and follow its evaluation protocols for testing. The dataset contains 46 scenes for training, 8 scenes for validation and 8 scenes for testing. The original 3DMatch [26] only considers point cloud pairs with >30% overlap. In addition to this benchmark (3DMatch), we follow [9] to include the collection of point cloud pairs with overlaps between 10% and 30% to form another benchmark (3DLoMatch).

Table 1: Raw data used in the 3DMatch [26] dataset and their licenses.

| Datasets | License |
|---|---|
| SUN3D [24] | CC BY-NC-SA 4.0 |
| 7-Scenes [17] | Non-commercial use only |
| RGB-D Scenes v.2 [11] | (License not stated) |
| Analysis-by-Synthesis [21] | CC BY-NC-SA 4.0 |
| BundleFusion [5] | CC BY-NC-SA 4.0 |
| Halbel et al. [8] | CC BY-NC-SA 4.0 |

**OdometryKITTI**  KITTI [7] is published under the NonCommercial-ShareAlike 3.0 License. It contains 11 sequences scanned by a Velodyne HDL-64 3D laser scanner in outdoor driving scenarios.

Following [1, 3, 9], we use sequences 0-5 for training, 6-7 for validation and 8-10 for testing. In line with [1, 3, 9], we further refine the provided ground truth poses using ICP [2] since them provided by GPS are noisy, and we only pick point cloud pairs with at least 10m intervals for evaluation.

## A.5 Additional ablation studies

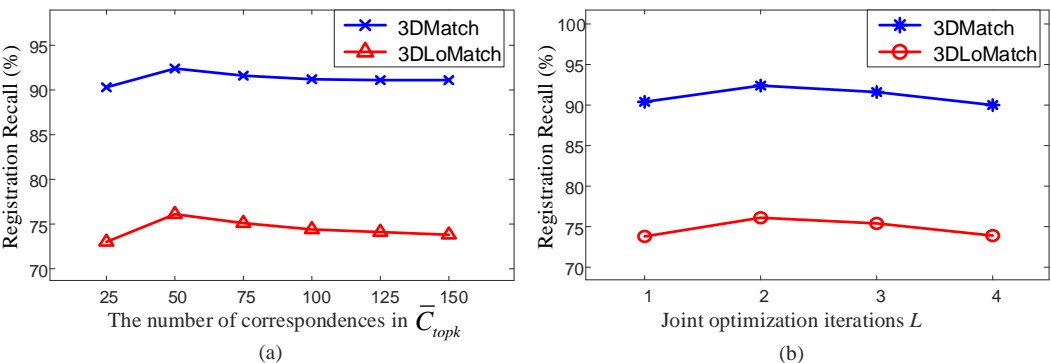

Figure 1: Additional ablation studies.

**The number of correspondences in $\bar{\mathbf{C}}_{topk}$.** We evaluate the impact of the number of correspondences in $\bar{\mathbf{C}}_{topk}$, which is used to generate the virtual correspondence. We vary the number of correspondences in $\bar{\mathbf{C}}_{topk}$ and calculate the registration recall. As shown in Fig. 1 (a), the registration recall improves with the increase of the number of correspondences in $\bar{\mathbf{C}}_{topk}$, i.e., from 25 to 50, and reaches saturation after 50 on 3DMatch dataset. On 3DLoMatch the registration recall reaches its peak at 50 and decreases after 50, because the proportion of false correspondences in $\bar{\mathbf{C}}_{topk}$ gradually rises. We choose the number of correspondences in $\bar{\mathbf{C}}_{topk}$ is 50 in our method to balance accuracy and speed.

**Iterative joint optimization.** Since we realize the joint optimization in an iterative fashion, by repeating the joint optimization operation different times, we analyze the effect of the number of joint optimization iterations on the final point cloud registration. As shown in Fig. 1 (b), iterative joint optimization yields an increase of registration recall, which peaks at the second iteration. After the second iteration the registration recall decreases as the number of iterations increases, a possible reason is that as the network deepens, the learning of network parameters becomes difficult. We choose $L = 2$ in our network, only repeating the joint optimization two times is one of the keys to our fast inference speed.

**The type of position encoding.** We attempt two types of position encoding for encoding the point-wise geometric position features: MLP position encoding and sinusoidal position encoding. For MLP position encoding, we use a 5-layer MLP with 32-64-128-256-256 channels. For sinusoidal position encoding, we extend the sinusoidal position encoding in [22] to 3D continuous coordinates. Sinusoidal position encoding has a slightly lower performance in registration accuracy than MLP type. Thus, we select the MLP position encoding.

Table 2: Influence of different position encodings.

| Pos. | 3DMatch | | | 3DLoMatch | | |
|---|---|---|---|---|---|---|
| | RR(%) | FMR(%) | IR(%) | RR(%) | FMR(%) | IR(%) |
| MLP | 92.4 | 98.1 | 62.3 | 76.1 | 84.6 | 27.5 |
| Sine | 91.6 | 98.2 | 62.7 | 75.2 | 85.0 | 26.7 |

**Absolute *or* relative position encoding.** We provide an ablation study to show the comparison between absolute and relative position encoding. Results can be found in Tab. 3, where we report the results of ours, absolute position encoding and centroid based relative position encoding. For absolute position encoding, we utilize the raw coordinates of point clouds to encode geometric position features. For centroid based position encoding, we use the centroids as reference points for

relative position encoding. The results show that our relative position encoding achieves the best performance.

Table 3: Comparison between absolute and relative position encoding.

| Model | 3DMatch | | | 3DLoMatch | | |
|---|---|---|---|---|---|---|
| | RR(%) | FMR(%) | IR(%) | RR(%) | FMR(%) | IR(%) |
| absolute | 89.5 | 97.3 | 56.6 | 68.4 | 82.2 | 23.7 |
| centroid based | 89.8 | 97.2 | 57.1 | 69.8 | 82.7 | 24.2 |
| ours | 92.4 | 98.1 | 62.3 | 76.1 | 84.6 | 27.5 |

## A.6 Additional experimental results

We present the results of *Relative Rotation Error* (RRE) and *Relative Translation Error* (RTE) on 3DMatch and 3DLoMatch in Tab. 4. Since the results of some failure cases may produce extremely large errors of translation and rotation, following [1, 3, 9, 14], we report the mean RRE and RTE for the successfully registered point cloud pairs. This measurement strategy makes methods with high registration recall more likely to have large mean errors because they include more difficult data in the calculation of mean errors. As shown in 4, our method still achieves competitive performances when compared with the closest competitor Gentransformer.

Table 4: Detailed results on the 3DMatch and 3DLoMatch datasets.

| | 3DMatch | | | | | | | | | | 3DLoMatch | | | | | | | | | |
|---|---|---|---|---|---|---|---|---|---|---|---|---|---|---|---|---|---|---|---|---|
| Methods | RRE ($^\circ$) | | | | | RTE (m) | | | | | RRE ($^\circ$) | | | | | RTE (m) | | | | |
| # Samples | 5000 | 2500 | 1000 | 500 | 250 | 5000 | 2500 | 1000 | 500 | 250 | 5000 | 2500 | 1000 | 500 | 250 | 5000 | 2500 | 1000 | 500 | 250 |
| Geotrans | 1.871 | 1.924 | 1.929 | 1.959 | 2.047 | 0.065 | 0.067 | 0.066 | 0.066 | 0.068 | 2.954 | 3.007 | 3.129 | 3.089 | 3.187 | 0.090 | 0.091 | 0.093 | 0.093 | 0.093 |
| Ours | 1.859 | 1.895 | 1.940 | 1.981 | 2.023 | 0.064 | 0.064 | 0.067 | 0.070 | 0.068 | 3.040 | 3.026 | 3.117 | 3.073 | 3.203 | 0.092 | 0.092 | 0.092 | 0.093 | 0.095 |

## A.7 Broader Impact

We present a one-inlier based position encoding for point cloud registration. It is very efficient, and competitive in term of computing overhead and inference speed. It provides a new perspective for real-time accurate point cloud registration tasks. Our method is most likely to be applied to autonomous driving and scene reconstruction. It could provide fast and accurate localization and scene perception for autonomous vehicles. Besides, real-time scene reconstruction is the key for intelligent embodied devices to realize scene understanding. Our method provide a new direction for it. Additionally, we hope to test the effectiveness of our approach for other fields involving registration task, including medical imaging and high-energy particle physics, etc. Point cloud registration is a fundamental task in computer vision and computer graphics. Therefore, potential negative societal impacts may occur when our method is applied to real scenarios. A possible case is the autonomous driving scenarios: our algorithms may fail in the presence of complex environments resulting in wrong driving decisions.

## A.8 Qualitative Results

We show qualitative results on 3DMatch/3DLoMatch and KITTI in Fig. 2 and Fig. 3, respectively. We also provide the visualization of failure cases of our method in Fig. 4. We observe that one common failure case happens when the overlapping region is composed of repetitive structures or textureless structures (e.g., wall, floor), resulting in failing to find reliable correspondences. The potential solutions include rejecting outliers as a post-processing, or applying point cloud completion based on the scene context.

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

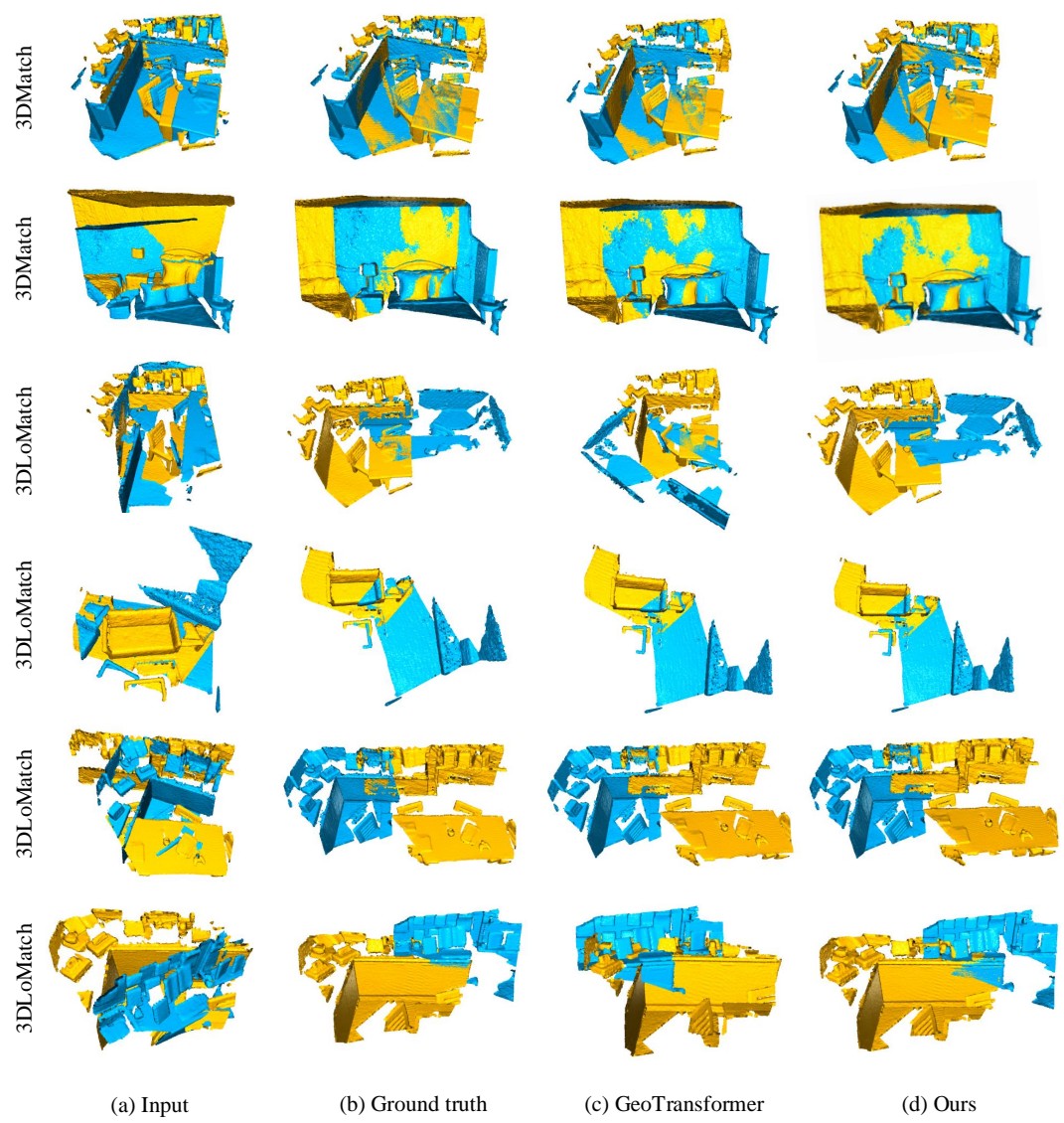

(a) Input          (b) Ground truth          (c) GeoTransformer          (d) Ours

Figure 2: Qualitative registration results on 3DMatch and 3DLoMatch.

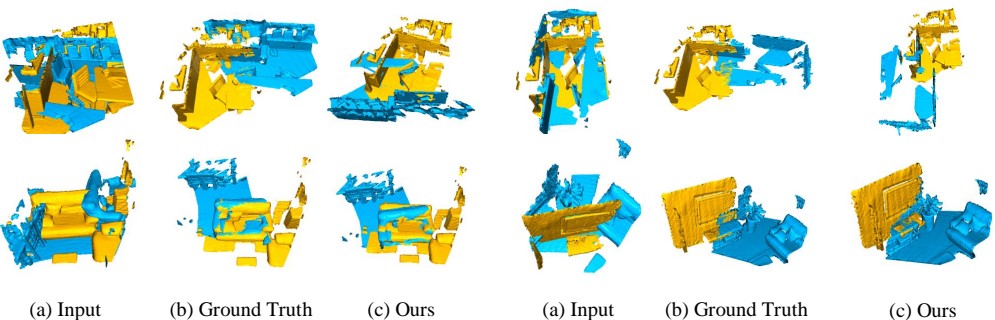

| (a) Source | (b) Target | (c) Ground truth | (d) Ours |

Figure 3: Qualitative registration results on KITTI.

| (a) Input | (b) Ground Truth | (c) Ours | (a) Input | (b) Ground Truth | (c) Ours |

Figure 4: Failure cases on 3DLoMatch.

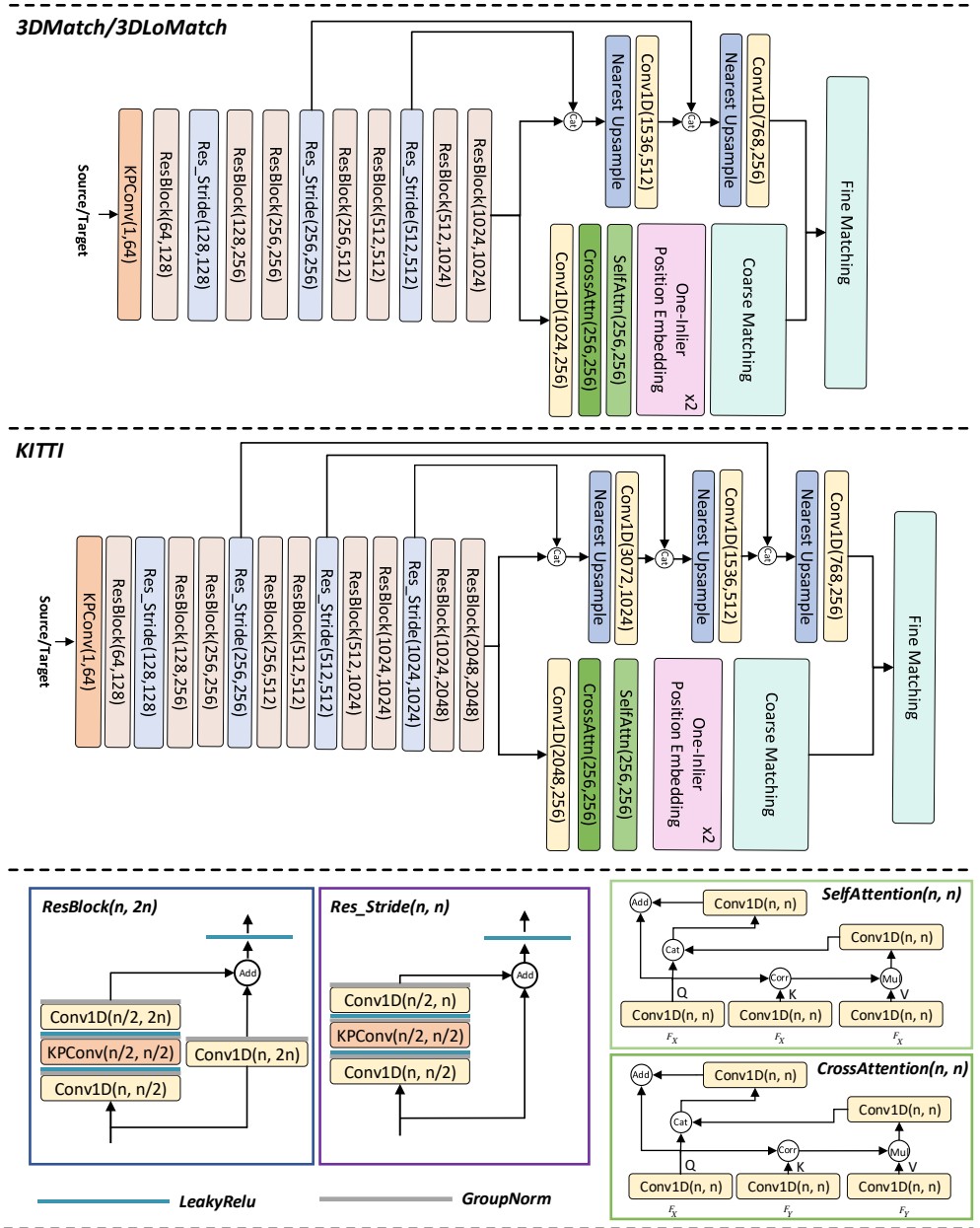

Figure 5: Network architecture for 3DMatch/3DLoMatch and KITTI.