# OpenReview forum: "One-Inlier is First: Towards Efficient Position Encoding for Point Cloud Registration"
_NeurIPS.cc/2022/Conference — NeurIPS 2022 Accept_

### Official Review · Reviewer_2PZE · 2022-07-08

**Rating:** 5
**Confidence:** 5
**Soundness:** 3 good
**Presentation:** 3 good
**Contribution:** 2 fair

**Summary:**

This paper studies correspondence-based rigid point cloud registration problem and proposes a new position encoding method for point clouds to establish better correspondences. A virtual corresponding point pair, which is regarded as the one inlier, is first constructed from a set of real correspondences, and then the two point clouds are normalized by using the two points in the correspondence as reference and the position embedding is extracted from the normalized point clouds. Point features and position embedding are added together for establishing correspondences. Based on the above method of establishing correspondences, an iterative strategy is adopted to align two point clouds gradually. Experiments are conducted on 3DMatch, 3DLoMatch and KITTI, and sota registration recall are achieved on these datasets.

**Questions:**

1. In section 4.1, especially for the experiments on 3DLoMatch, the proposed method consistently has higher RR but lower FMR and IR. I agree that FMR and IR are not total positively-correlated to RR, but more analysis is needed to show why the proposed method has higher RR with a far lower IR, and how this is related to the proposed position encoding strategy.
2. For the “Progressive alignment”, how many iterations are performed?
3. For Fig. 3, the inlier ratio of every iteration should be given, instead of only the initialization and the final optimized one.
4. In Table 2, what constitutes the “Pose” Time? Is it the runtime of the RANSAC 50k? If so, why is the proposed method much faster than GeoTransformer with lower IR?
5. What’s the number of sample correspondences in the runtime experiment in Table 2?
6. The quality of the “one-inlier” for position encoding  should be given in each iteration.
7. RRE and RTE should be reported for the experiments on 3DMatch and 3DLoMatch since the objective of this paper is registration.


**Limitations:**

The authors talk about the limitation of the proposed method in supplementary material. I suggest address the issue about if the proposed method is applicable to registration of objects, such as models in ModelNet40.

**Strengths And Weaknesses:**

Strengths:
1.The proposed one-inlier based position encoding method is novel. The method is straightforward and effective.
2.Experiments are conducted on three typical datasets and comparison are done to latest methods. High performance is achieved.

Weakness:
1. A good position encoding should be helpful to find more and better correspondences and a better registration is only a by-product of better correspondences, but the experimental results show that the IR and FMR of the proposed method are far lower than that of GeoTrans. Analysis are needed about this issue.
2. The last two contributions are not significant. Especially, iterative optimization is widely used in recent learning-based registration methods, such as PRNet, RPM-Net.
3. The proposed position-encoding idea is not fully analyzed. See question No.1.

Minor issue:
1. The part “one-inlier is enough” in title is a little misleading.
2. In Fig.2, it should be the Matched Correspondences instead of {R, t}, that are transferred from the Inlier Learning module to the Progressive Alignment module.

---

> ### Author Response · Authors · 2022-08-02
> **Response to Reviewer 2PZE**
>
> 1. **Q**: A good position encoding should be helpful to find more and better correspondences and a better registration is only a by-product of better correspondences, but the experimental results show that the IR and FMR of the proposed method are far lower than that of GeoTrans. Analysis are needed about this issue.
> **A**: Question 1 and Weakness 1 are the similar issues, so we answer them here together. As pointed in [17, 2, 7, 47], registration recall is the more important metric than IR and FMR in point cloud registration task, because the final goal of point cloud registration is to estimate the rigid transformation. Although the inlier rate of our method is lower than Geotransformer, our method can consistently achieve the best registration recall in all settings. We analyze the reason why our method can get better registration recall while the inlier ratio is not as good as Geotransformer in the Part 2 of the General Response section.
> 2. **Q**: The last two contributions are not significant. Especially, iterative optimization is widely used in recent learning-based registration methods, such as PRNet, RPM-Net.
> **A**: Indeed, iterative optimization has been widely used in recent learning-based registration methods. Some learning-based registration methods do correspondence establishment and the final transformation estimation in an iterative manner, similar to the classical ICP algorithm, such as PRNet, RPM-Net. However, our iterative optimization aims to find accurate reference points and progressively accurate position encoding, not for the final correspondence establishment and transformation estimation. Our method jointly optimizes the reference points and position encoding with point-wise feature encoding as the agent several times. In the process, the features of the point clouds are continuously optimized. Finally, the optimized point cloud features are utilized for the final correspondence establishment.
> 3. **Q**: The part “one-inlier is enough” in title is a little misleading.
> **A**: The explanation and the revised proposal of the title are presented in the Part 1 of the General Response section.
> 4. **Q**: In Fig.2, it should be the Matched Correspondences instead of {R, t}, that are transferred from the Inlier Learning module to the Progressive Alignment module.
> **A**: We will revise this part and make sure Fig.2 is clear in the revised version.
> 5. **Q**: For the “Progressive alignment”, how many iterations are performed?
> **A**: We repeat the “progressive alignment” 2 times. We have clarified it in the supplementary material.
> 6. **Q**: For Fig. 3, the inlier ratio of every iteration should be given, instead of only the initialization and the final optimized one.
> **A**: We perform the iterative joint optimization only two times. The initialization one and the final optimized one means the inlier ratio of  $\bar{\textbf{C}}_{topk}$ in the first and second iterative joint optimization, respectively.
> 7. **Q**: In Table 2, what constitutes the “Pose” Time? Is it the runtime of the RANSAC 50k? If so, why is the proposed method much faster than GeoTransformer with lower IR?
> **A**: We test the runtime with RANSAC 50k. GeoTransformer utilizes the RANSAC with 50k iterations and 50k validation, while our method uses the RANSAC with 50k iterations and 10k validation. Too much validation times cause that GeoTransformer inferences too slow and it is difficult to apply it to real-world scenarios.
> 8. **Q**: What’s the number of sample correspondences in the runtime experiment in Table 2?
> **A**: The number of sample correspondences is 5000 in the runtime experiment. We will add a description in the revised version.
> 9. **Q**: The quality of the “one-inlier” for position encoding should be given in each iteration.
> **A**: We represent the quality of the “one-inlier” by the inlier ratio of $\bar{\textbf{C}}_{topk}$ in Fig. 3. We perform the iterative joint optimization two times, so we show the statistical results in the two iterations (represented as “initialization” and “optimized” respectively).
> 10. **Q**: RRE and RTE should be reported for the experiments on 3DMatch and 3DLoMatch since the objective of this paper is registration.
> **A**: We provide the RRE and RTE of our method and the closest competitor Gentransformer in Tab.3 of supplementary material.
> 11. **Q**: I suggest address the issue about if the proposed method is applicable to registration of objects, such as models in ModelNet40.
> **A**: Limited by the time of rebuttal, we do not have enough time to complete the experiment at this time. We will conduct experiments on object benchmark (e.g., ModelNet40) to verify the applicability of the proposed method to the registration of objects in the revised version.

---

> > ### Comment · Reviewer_2PZE · 2022-08-07
> > **Further clarification about question 7 and 9**
> >
> > I appreciate the detailed response from the authors. Some further clarification is desired about question 7 and 9.
> >
> > In reply to question 7, you mentioned "GeoTransformer utilizes the RANSAC with 50k iterations and 50k validation, while our method uses the RANSAC with 50k iterations and 10k validation". Do you also used RANSAC with "10k validation" in your experiments for Table 1?
> >
> > About the quality of the “one-inlier” in question 9. To be effective for position encoding of other points, I think the two points in the one-inlier correspondence should be true corresponding points. By quality, what I want to know is how accurate this correspondence is. Do you calculate the distance between the two points under ground-truth transformation?

---

> > > ### Author Response · Authors · 2022-08-08
> > > **Further response about question 7 and 9**
> > >
> > > 1. **Q**: Do you also used RANSAC with "10k validation" in your experiments for Table 1?
> > > **A**: Yes. In Table 1, we also used RANSAC with "10k validation" in our experiments.
> > > 2. **Q**: To be effective for position encoding of other points, I think the two points in the one-inlier correspondence should be true corresponding points. By quality, what I want to know is how accurate this correspondence is. Do you calculate the distance between the two points under ground-truth transformation?
> > > **A**: Our method employs a coarse-to-fine manner to find correspondences and aims at finding coarse correspondences (patch correspondences) in the coarse stage. Our position encoding is designed for establishing coarse correspondences. The higher the precision of the “one inlier”, the more accurate our position encoding will be. The “one inlier” is calculated by an averaging operation of $\bar{\textbf{C}}_{topk}$ (because of formatting issues it is represented as C-topk in the following part), thus we represent the precision of the “one-inlier” by the inlier ratio of C-topk in Fig. 3. Following GeoTransformer [29], we also consider a coarse correspondence is correct if the overlap of the corresponding two patches is greater than a predefined threshold (we set the threshold to be 0 as GeoTransformer [29]) under ground-truth transformation. Otherwise, it will be regarded as a wrong coarse correspondence. According to this way, we determine if each coarse correspondence is an inlier and count the inlier ratio of C-topk in Fig. 3. For the overlap of two patches, we calculate the alignment distance of each point in the patches, i.e., calculating the distance between the two points under ground-truth transformation. If the alignment distance of a point in one patch is lower than a predefined threshold (we set the threshold to be 0.05 as GeoTransformer [29]), it will be considered as being in the overlapping area. In Fig. 3, we count the inlier ratio of C-topk, the experimental results show that the inlier ratio increases after optimization. Thus the precision of the “one-inlier” is improved.

---

### Official Review · Reviewer_VUqY · 2022-07-11

**Rating:** 6
**Confidence:** 4
**Soundness:** 3 good
**Presentation:** 3 good
**Contribution:** 3 good

**Summary:**

This paper proposes a progressive point cloud registration method that estimates an initial transformation between two point clouds before registration estimation. The authors claim that one inlier is enough to estimate a good enough initial transformation between two point clouds, which boosts the downstream point cloud registration process, including positional encoding and progressive alignment. The main contributions of this paper are a) the one inlier finding process which is light-weight, b) the process can be jointly optimized with other steps, and c) the progressive alignment approach reduces the dependency on initialization.

**Questions:**

#### Q1 (ad W2): Inaccurate or Unrelated Citations
line 32: "... the straightforward position encoding is not a good idea [23]."
In [23] it shows up to 3.8% improvement on 4DLoMatch and 1.5% higher RR on 3DMatch and 2.3% on 3DLoMatch.
This does not imply straightforward position encoding is not a good idea.
To make this statement true, one should show the comparison with/without direct position encoding, which is not provided in the paper.

line 51: "This shows that only one inlier is enough to preserve the spatial consistency [2, 7] ..."
I am not sure why those two papers are cited here. Do they mention anything related to this statement?

#### Q2 (ad W3): Claims Without Proof:
line 286: " This is because ... both consider the position information ..."
To justify this statement, one should provide an ablation study on a model with only the difference with and without positional encoding. However, this is not provided in the evaluation section.

line 342: "our method is scene-agnostic and maintains good registration accuracy in strongly differing scenarios."
Without a description of how the network was trained in those experiments, it is impossible to know whether the written sentence is true.

#### Q3: Additional questions
- In Figure 3, the inlier ratio for 3DLoMatch. Why does the 0.0 percentage increase after optimization?

**Limitations:**

The authors do not mention limitations in the main paper. No text in the main paper indicates a limitation section in the supplementary material.

**Strengths And Weaknesses:**

### Strengths:
#### S1. Core Idea.
The idea is novel and outperforms other methods in two public benchmarks.
#### S2. Impact on Applications.
The method is lightweight and can potentially be applied in many other pipelines / with other methods, which benefits the community.
#### S3. The paper is well written and easy to follow.

### Weaknesses:
#### W1. Limited theoretical / technical Novelty
Although the paper is novel regarding the idea, the theoretical and technical novelty is limited.
#### W2. Inaccurate Claims
This paper has made some claims with some cited papers that are inaccurate or unrelated (details below).
#### W3. Missing Justification
Some claims are made without proper proof (see details below).
#### W4. Minor Typos / Writing
* line 134: "C is ..." -> "where C is ..."
* Eq 5: "..., M," -> "..., M."

---

> ### Author Response · Authors · 2022-08-02
> **Response to Reviewer VUqY**
>
> 1. **Q**: line 32: "... the straightforward position encoding is not a good idea [23]." In [23] it shows up to 3.8% improvement on 4DLoMatch and 1.5% higher RR on 3DMatch and 2.3% on 3DLoMatch. This does not imply straightforward position encoding is not a good idea. To make this statement true, one should show the comparison with/without direct position encoding, which is not provided in the paper.
> **A**: For the “straightforward position encoding”, it means the absolute position encoding. [23] is cited here because it also indicates that the straightforward position encoding is not a good idea. Thus, it [23] proposes a relative position encoding. However, it is still not enough to just satisfy the relative position encoding. An arbitrary reference point would introduce high uncertainty, because the two reference points of point clouds are hardly guaranteed to be correlated. Therefore, we propose the one-inlier based position encoding to compensate the positional differences caused by reference frames. In fact, we have presented the experiments about replacing the one-inlier based position encoding with the centroid based position encoding in our ablation studies, i.e., “w/o associated reference points”. Besides, we also provide an experiment about replacing our relative position encoding module with the absolute position encoding. Results can be found in the following table, where we report the results of ours, absolute position encoding and centroid based position encoding. The results show that our position encoding achieves the best performance.
> **3DMatch:**
> |Methods|RR(%)|FMR(%)|IR(%)|
> |:--------------------|:-----:|:-----:|:-----:|
> |absolute|89.5|97.3|56.6|
> |centroid based|89.8|97.2|57.1|
> |ours|92.4|98.1|62.3|
> **3DLoMatch:**
> |absolute|68.4|82.2|23.7|
> |centroid based|69.8|82.7|24.2|
> |ours|76.1|84.6|27.5|
> 2. **Q**: line 51: "This shows that only one inlier is enough to preserve the spatial consistency [2, 7] ..." I am not sure why those two papers are cited here. Do they mention anything related to this statement?
> **A**: Those two papers both introduce the spatial consistency for finding good correspondences in point cloud registration. They verify that preserving the spatial consistency is beneficial to point cloud registration. We cite them here because the proposed one-inlier based position encoding method also tries to utilize the spatial consistency constraint.
> 3. **Q**: line 286: "This is because ... both consider the position information ..." To justify this statement, one should provide an ablation study on a model with only the difference with and without positional encoding. However, this is not provided in the evaluation section.
> **A**: In the following table, we present an ablation study on our model. We remove the progressive alignment module in all the following experiments. “iterative position encoding” means that we combine the proposed position encoding with the joint optimization. We can see that the performance of our method benefits from the position encoding.
> **3DMatch:**
> |Methods|RR(%)|FMR(%)|IR(%)|
> |:--------------------|:-----:|:-----:|:-----:|
> |w/o position encoding|88.9|96.7|45.0|
> |with position encoding|91.1|97.5|58.9|
> |iterative position encoding|91.4|98.2|61.8|
> **3DLoMatch:**
> |w/o position encoding|69.1|81.3|22.4|
> |with position encoding|72.6|83.6|25.7|
> |iterative position encoding|74.2|84.1|27.1|
> 4. **Q**: line 342: "our method is scene-agnostic and maintains good registration accuracy in strongly differing scenarios." Without a description of how the network was trained in those experiments, it is impossible to know whether the written sentence is true.
> **A**: “our method is scene-agnostic” means that our method performs well on differing scenarios, including indoor and outdoor scenes. We provide the extensive description about how to train the network on different datasets in the supplementary material, including the network architecture details, implementation details and datasets.
> 5. **Q**: In Figure 3, the inlier ratio for 3DLoMatch. Why does the 0.0 percentage increase after optimization?
> **A**: In fact, the “0.0 percentage” you said corresponds to the interval [0, 0.025]. For extreme low-overlap scenarios, the initial inlier ratio of $\bar{\textbf{C}}_{topk}$ may be very low, which would cause the produced virtual correspondence not to be a truth correspondence and introduce false position encoding for the subsequent feature reconstruction. Finally, due to the introduction of extremely wrong position encoding, the optimized inlier ratio decreases and the number of scenes with inlier ratios close to 0 increases after optimization.
> 6. **Q**: The authors do not mention limitations in the main paper. No text in the main paper indicates a limitation section in the supplementary material.
> **A**: We present the limitations in line 128-135 of supplementary material and we will either mention them in or move them to the main body in the revised version.

---

> > ### Comment · Reviewer_VUqY · 2022-08-09
> > **Additional Evidence on Positional Encoding**
> >
> > Thanks for providing such a detailed answer!
> >
> > Especially the additional experimental evidence on the positional encodings helps to overcome the doubts verbalized in the review. The additional ablation (absolute, 1) and the performance analysis (3) would be great to be included in the paper / supplementary material as this question may arise and it backs up the claims made in the text.
> > The explanation in (5) equally prevents others from misunderstanding - would be good to add this comment.
> >
> > The new title suggestion is also more specific and fitting.

---

> > > ### Author Response · Authors · 2022-08-09
> > > **Thanks for your feedback!**
> > >
> > > Thank you for your constructive suggestions and helps about improving this paper! We will add the suggested experiments and explanations in the revised version.

---

### Official Review · Reviewer_fXaq · 2022-07-12

**Rating:** 6
**Confidence:** 5
**Soundness:** 3 good
**Presentation:** 3 good
**Contribution:** 3 good

**Summary:**

This paper proposes a position encoding method by using optimal transport and use it to normalize each point. The benefit is that the  optimal transport based position encoding can mitigate the feature ambiguity. Then, an iterative optimization process is introduced to establish the correspondence and transformation matrix.

**Questions:**

Why the inlier ratio is relatively low?

**Strengths And Weaknesses:**

Strength:
1. The idea is reasonable and the overall performance is good.
2. The presentation is easy to understand.

Weakness:
1. I am a bit worried about the robustness of the proposed algorithm since the inlier ratio is significantly dropped.

---

> ### Author Response · Authors · 2022-08-02
> **Response to Reviewer fXaq**
>
> 1. **Q**: I am a bit worried about the robustness of the proposed algorithm since the inlier ratio is significantly dropped.
> **A**: It is true that our method performs worse than GeoTransformer in terms of inlier ratio, but our method achieves the highest registration recall (i.e., the most important metric in point cloud registration task) and performs consistently well under different numbers of sampled correspondences. In fact, inlier ratio does not totally reflect the robustness of the registration. Because usually the inlier ratio is counted by averaging all the point cloud pairs. For some easy-to-handle scenarios, if the inlier ratio reaches a certain extent, the higher inlier ratio will have no more contributions to the final registration results, but the mean inlier ratio will rise. We think that the inlier ratio in the challenging scenes better reflects the robustness of the algorithm rather than the mean inlier ratio. To illustrate the issue more intuitively, we count the scene frequency of different inlier ratios for Geotransformer and our method, and present the curve charts of scene distributions in Fig.2 of supplementary material. Specifically, we count the number of scenes with different inlier ratios and present them in frequency form. Besides, we also count the scene registration recall of different inlier ratios and provide the curve charts in Fig.3 of supplementary material. Here we take the results on 3DLoMatch dataset as an example, and the results show that: 1) the inlier ratio of our method is between 0.05 and 0.6 in most scenarios. For Geotransformer, it has more scenarios where the inlier ratio is between 0.5 and 0.9. However, RANSAC has certain anti-noise capability. When combined with the Fig. 3 of the supplementary material we can see that our relatively lower inlier rate is adequate to achieve the high registration recall. High inlier ratio is not a requisite for the high registration recall. 2) Besides, it is worth noting that our method has fewer scenarios than Geotransformer in extremely low inlier ratio, which proves that our method is more robust for extremely challenging scenes. For most scenarios, our method is stable. Although it can’t provide inlier ratio as high as Geotransformer, it generally has the ability to provide enough inliers and still has more consistent performance than Geotransformer. In fact, we can get a similar conclusion by analyzing the experimental results on 3DMatch dataset.
> In addition, our method can achieve consistently competitive performance in both indoor and outdoor datasets, even the low overlap scenes. It also proves that our method has strong robustness and adaptability to different scenarios.
> 2. **Q**: Why the inlier ratio is relatively low?
> **A**: Please refer to the Part 2 in General Response section, we present a detailed analysis about why the inlier ratio of our method is relatively lower than Geotransformer and the relationship between the registration results and the established correspondences.
> 3. **Q**: Limitations.
> **A**: We present the limitations in line 128-135 of supplementary material and we will either mention them in or move them to the main body in the revised version.

---

### Official Review · Reviewer_dDv9 · 2022-07-15

**Rating:** 5
**Confidence:** 5
**Soundness:** 3 good
**Presentation:** 3 good
**Contribution:** 2 fair

**Summary:**

This paper presents a learning-based algorithm that establishes reliable correspondences for registering a pair of point clouds.
Point cloud registration has been a long studied problem in robotics and computer vision, and there exist many papers that propose learning-based methods to perform feature learning and establish correspondences.
In my view, there are two unique ideas about this paper that differ it from previous papers:
- The proposed pipeline predicts an "anchor" point for each point cloud (in the paper this is called "one inlier") and uses the "anchor" point to "normalize" the rest of the point cloud (in the paper, the normalization is basically subtracting the anchor point from each other point in the cloud). After normalization, an additional feature learning step is performed to augment the coarse features (learned by KPConv).
- The proposed pipeline does correspondence learning and transformation estimation in an iterative manner, similar to the classical ICP algorithm. In the classical ICP, correspondences are established via nearest neighbor search. But in this paper, correspondences are established from learning, using the features learned from the "anchoring" idea mentioned above.

This paper then tests the proposed pipeline in the 3DMatch and 3DLoMatch dataset, and demonstrates the proposed pipeline delivers similar or better performance compared to several previous learned features for point cloud registration.

**Questions:**

Since you know the groundtruth pose, could you add into the loss function a pose error and see how it affects the performance?

**Ethics Review Area:**

["I don’t know"]

**Limitations:**

Limitations not mentioned.

**Strengths And Weaknesses:**

Strengths:
+ The paper is overall well written and easy to understand.
+ The idea of predicting or voting an "anchor" point and using it to normalize the point cloud seems to be relatively new (though I feel this idea is similar to Hough voting).
+ Experiments are well conducted and convincing.

Weaknesses:
- I feel the paper is a little bit out of fashion/date, given that the community has designed many **supervised** learned point cloud feature descriptors and moved to studying **unsupervised** or "self-supervised" feature descriptors. The proposed descriptor, though outperforming others, is still supervised and requires mining ground-truth correspondences for supervision. [Ref1] and others, for example, have already shown the possibility to learn features in a self-supervised way. Therefore, I think the paper is making an incremental contribution rather than a major contribution.
- The title of this paper, to me, is deceiving and jargoning. One inlier correspondence is NOT enough to register a pair of point clouds, and at least THREE inlier correspondences (with noncollinear points) are required to define a unique rigid transformation (see [Ref2]). What the paper really means is to predict a "anchor" point (sort of like the centroid of the point cloud) and use the anchor point to subtract all other points. I think the name of the paper needs to be better worded.


[Ref1] Yang H, Dong W, Carlone L, Koltun V. Self-supervised geometric perception. In Proceedings of the IEEE/CVF Conference on Computer Vision and Pattern Recognition 2021 (pp. 14350-14361).
[Ref2] Horn BK. Closed-form solution of absolute orientation using unit quaternions. Josa a. 1987 Apr 1;4(4):629-42.

---

> ### Author Response · Authors · 2022-08-02
> **Response to Reviewer dDv9**
>
> 1. **Q**: I feel the paper is a little bit out of fashion/date, given that the community has designed many supervised learned point cloud feature descriptors and moved to studying unsupervised or "self-supervised" feature descriptors. The proposed descriptor, though outperforming others, is still supervised and requires mining ground-truth correspondences for supervision. [Ref1] and others, for example, have already shown the possibility to learn features in a self-supervised way. Therefore, I think the paper is making an incremental contribution rather than a major contribution.
> **A**: Recently several unsupervised or self-supervised works have been proposed to solve the point cloud registration problem and achieved good performances. The supervised point cloud registration task still receives great attention and many works have emerged, such as Predator, CoFiNet, REGTR, Geotransformer and so on. The unsupervised or self-supervised methods do not require ground-truth annotations and have good generalization ability, but the registration accuracy is limited and it is currently difficult to exceed the supervised methods. For supervised methods, they could achieve the state-of-the-art performances, but rely on a sufficient number of ground-truth labels. Whether supervised or unsupervised methods, they both are worth further exploration. The proposed method belongs to the category of supervised methods and our main contributions are: 1) we propose an efficient one-inlier based position encoding for point cloud registration and achieve competitive performance with the latest state-of-the-art approaches. 2) Our method is lightweight and very efficient, and the GPU memory usage is only about 40% of GeoTransformer’s. Besides, it is fast and our registration speed is 3.5 times faster than GeoTransformer. Thus, our method has the potential to be applied in real scenarios.
>
> 2. **Q**: The title of this paper, to me, is deceiving and jargoning. One inlier correspondence is NOT enough to register a pair of point clouds, and at least THREE inlier correspondences (with noncollinear points) are required to define a unique rigid transformation (see [Ref2]). What the paper really means is to predict a "anchor" point (sort of like the centroid of the point cloud) and use the anchor point to subtract all other points. I think the name of the paper needs to be better worded.
> **A**: The explanation and the revised proposal of the title are presented in the Part 1 of the General Response section.
> 3. **Q**: Since you know the groundtruth pose, could you add into the loss function a pose error and see how it affects the performance?
> **A**: We have attempted to add a loss function term about the error of the estimated pose in each progressive alignment process. Results can be found in the following table. We can observe that the “pose loss term” has limited effect on the performance. We analyze that the “pose loss term” plays the similar role as coarse correspondence loss and fine correspondence loss in the main body in the network training.
> **3DMatch:**
> |Methods|RR(%)|FMR(%)|IR(%)|
> |:--------------------|:-----:|:-----:|:-----:|
> |with pose loss term|91.8|97.0|62.5|
> |w/o pose loss term|92.4|98.1|62.3|
> **3DLoMatch:**
> |with pose loss term|75.6|84.4|26.0|
> |w/o pose loss term|76.1|84.6|27.5|
>
> 4. **Q**: Limitations not mentioned.
> **A**: Due to the limitation of pages, we introduce the limitations in line 128-135 of supplementary material and we will either mention them in or move them to the main body in the revised version.

---

### Author Response · Authors · 2022-08-02
**General Response**

Thank all the reviewers for their careful review. For the grammar mistakes, we will carefully proofread them later.
1. **About the title may be a little misleading (To Reviewer dDv9 Q2 and Reviewer 2PZE Q3).**
“One inlier” means that we find a virtual correspondence, then the two points corresponding to the correspondence are considered as the reference points. Next, the reference points are utilized to further encode point-wise geometric position features, and generate more discriminative features for the point cloud. We expect the found virtual correspondence to be an inlier, and design iterative optimization strategies to realize this. Once we find an inlier, it is utilized to efficiently perform the proposed position encoding. We also realize that the title may not accurately express our core ideas and is a little misleading, because after finding an inlier, we leverage the inlier to perform the subsequent operations, including: position encoding for feature reconstruction and several times joint optimization. Thus, we consider revising the title to: “One Inlier is First: Towards Efficient Position Encoding for Point Cloud Registration”, and emphasize that finding an inlier is an important prerequisite for our approach.
2. **The analysis about the relationship between the registration results and the established correspondences (To Reviewer fXaq Q2 and Reviewer 2PZE Q1).**
In order to analyze the reason why our method can get better registration recall while the inlier ratio is not as good as Geotransformer, we conduct an experiment: we count the scene frequency of different inlier ratios for Geotransformer and our method, and present the curve charts of scene distributions in Fig.2 of supplementary material. Specifically, we count the number of scenes with different inlier ratios on 3DMatch and 3DLoMatch datasets, and present them in frequency form. Besides, we also count the scene registration recall of different inlier ratios and provide the curve charts in Fig.3 of supplementary material.
Here we take the results on 3DLoMatch as an example, from the Fig. 2 of the supplementary material, we can observe that the inlier ratio of our method is about 0.05 to 0.6, while Geotransformer has more scenes with inlier ratio between 0.5 and 0.9 than ours. Geotransformer has higher inlier ratio in some scenarios, which causes the mean inlier ratio to be higher than ours. However, higher inlier ratio is not a necessary condition for high registration recall. For example, for the same scenario, suppose there are two different correspondence results with the same inlier ratio but different distributions of inliers, the correspondences with more uniform distribution of inliers may achieve better registration performance than the correspondences with more locally clustered inliers. This can be explained by the fact that too locally clustered distribution of inliers may cause degradation issues in model estimation. From Fig. 3 of the supplementary material we can observe that our method can achieve better registration recall than Geotransformer with the same inlier ratio (e.g, the interval [0.15, 0. 5]), which shows that the inliers of our method may be better distributed for model estimation. Meanwhile, it also explains why our method has lower inlier ratios, but achieves better registration results. Besides, when the inlier ratio is higher than 0.45, our registration recall is almost 1. This also proves that when the inlier ratio reaches a certain extent, higher inlier ratio has no more contributions for the final registration results. It is worth pointing out that our method has higher inlier ratio than other methods except for GeoTransformer. We speculate that our method constructs global positional features by first finding an inlier, which can introduce spatial consistency. So our method is possible to find matching pairs that cannot be found by local features alone, which may benefit the final registration task. Similarly, we can get this conclusion by analyzing the experimental results on 3DMatch dataset.

---

### Meta-Review · Area_Chair_Dep1 · 2022-08-23

**Recommendation:** Accept
**Confidence:** Certain

**Metareview:**

Thanks in large part to the rebuttal conversation, the reviewers converged to accept this paper.  The reviewers recognize the interest and value of the approach and careful empirical results, bolstered by additional results introduced during the discussion.

In preparing the camera-ready, the authors of this paper are encouraged to revisit the comments from reviewer 2PZE suggesting to verify whether the two anchor points are truly ‘inlier’ correspondence; this can be easily done by calculating the distance between the two anchor points under ground-truth transformation.  Also, please make the title change and any other edits promised in the rebuttal, especially discussion of drawbacks and avenues for future research.

**Award:**

No

---

### Decision · Program_Chairs · 2022-09-14

Accept